
**Spatially explicit assessment of water scarcity and potential mitigating solutions in a large water-limited basin: the Yellow River basin in China**

**Weibin Zhang[1,2], Xining Zhao[3,4], Xuerui Gao[4], Wei Liang[5], Junyi Li[5], and Baoqing Zhang[6*]**

[1]College of Water Resources and Architectural Engineering, Northwest A&F University, 712100 Yangling, Shaanxi Province, China.

[2]Key Laboratory of Agricultural Soil and Water Engineering in Arid and Semiarid Areas, Ministry of Education, Northwest A&F University, 712100 Yangling, Shaanxi Province, China.

[3]Institute of Soil and Water Conservation, Northwest A&F University, 712100 Yangling, Shaanxi Province, China.

[4]Institute of Soil and Water Conservation, Chinese Academy of Sciences and Ministry of Water Resources, 712100 Yangling, Shaanxi Province, China.

[5]School of Geography and Tourism, Shaanxi Normal University, 710119 Xi'an, 15    Shaanxi Province, China.

[6]Key Laboratory of Western China's Environmental Systems (Ministry of Education), College of Earth and Environmental Sciences, Lanzhou University, 730000 Lanzhou, Gansu Province, China.

**\*Corresponding author:**

Email: baoqzhang@lzu.edu.cn;

Tel (Fax): +86 (931) 8912404;

Address: NO. 222 Tianshui Road (South), Chengguan District, Lanzhou, Gansu Province, China



**Abstract:**

Comprehensive assessment of the long-term evolution of water scarcity and its driving factors is essential for designing effective water resource management strategies. However, the role of water withdrawal and water availability components in

determining water scarcity and potential mitigating measures in large water-scarce basins are poorly understood. Here, an integrated analytical framework was applied to the Yellow River basin (YRB), where the water crisis has been a core issue for sustainable development. Analysis of the water scarcity index (WSI) and other critical indicators, including frequency, duration, and exposed population, suggest that the

YRB experienced unfavorable changes in water crisis during 1965–2013. Irrigation dominated the increase in WSI in the northwest part of the basin before 2000, whereas climate change was primarily responsible for changes in the WSI in most sub-basins during the recent decade. Meanwhile, local water management and climate change adaptation were shown to be important in determining total water availability at the

sub-basin scale. Water demand in the 2030s is predicted to be 37.4 km$^3$ based on the trajectory of historical water use, worsening 28.7% and easing 12.5% of the total population, respectively. To meet all sectoral water needs, a 10 km$^3$ water deficit is projected. The potential improvements in irrigation efficiency could solve 26% of this deficit, thereby easing the pressure on external water transfer projects. In conclusion,

the integration of water supply and demand-oriented strategies is essential to effectively alleviate the water crisis in the YRB. Our results have vital implications for water resource management in basins facing similar water crises to that in the YRB.



## 1 Introduction

Water resources underpin human life, socio-economic development, and ecosystem health (Oki and Kanae, 2006; Han et al., 2023), but often have an uneven spatiotemporal distribution as well as a mismatch with water demands (Veldkamp et al., 2017; Wang et al., 2020; Scanlon et al., 2023). This problem has been worsened by an increased human water demand during the last few decades, driven by population growth,

improving living standards, and expansion of irrigated agriculture (McDonald et al., 2011; Wada et al., 2016b; Huang et al., 2018). Climate change poses an additional threat to already stressed water resources by adding uncertainty, especially in terms of changes in interannual and seasonal precipitation and temperature (Schewe et al., 2014; Rodell and Li, 2023). Water scarcity is challenging the stability and sustainable development

of human society in many regions of the world, especially in developing countries (Munia et al., 2020; Huang et al., 2021). Approximately one-third to half of the global population is currently experiencing water scarcity, with most water scarcity occurring in India and China (Mekonnen and Hoekstra, 2016; Qin, 2021). By 2030, half of the global population is predicted to experience severe water stress (UNEP, 2015).

Droughts and floods would further intensify water stress, with high-income areas not immune to these threats (Rodell and Li, 2023). Water scarcity results in many social and environmental issues, such as food production reduction, drinking water shortage, and ecosystem health degradation (Porkka et al., 2016; Wang et al., 2017; Long et al., 2020). Thus, it is essential to understand the evolution of water stress and the associated

driving factors; this is a prerequisite for designing effective water resource management strategies and achieving Sustainable Development Goal (SDG) target 6.

The Yellow River basin (YRB), the second-largest river basin in China, is known as the "Chinese cradle". It is responsible for 13% of national grain production but only possesses approximately 2% of national water resources (Zhuo et al., 2016). Meanwhile,

the reserves of coal and oil in this basin account for 70% and 50% of China's total, respectively (Ma et al., 2020b). The large-scale exploration of energy sources also comes at the cost of a large amount of water usage. Accompanying a boom in



agriculture and prosperous economy, the total water consumption (both surface and groundwater) in the YRB increased by 120% between the 1960s and 2009 (Zhuo et al.,

2016). In contrast, natural runoff significantly decreased during the period 1960–1990s, before slightly recovering in recent years (Tang et al., 2013). Owing to climate change-induced natural water availability and intensive human water usage, the YRB has been facing severe water scarcity (Xie et al., 2020; Niu et al., 2022; Zhang et al., 2023c). This problem has constrained the ecological protection and high-quality development

of the YRB, which was stressed as a major national strategy in 2019 by the Chinese government.

    Water scarcity in the YRB has been widely reported in global assessments, which have provided general overviews of water stress for both historical and future periods (Veldkamp et al., 2017; Greve et al., 2018; Qin et al., 2019). However, these estimates

were based on data having a coarse spatial resolution, such as a whole river basin or at $0.5° \times 0.5°$ level (Huang et al., 2021). The insights obtained at such spatial resolution might be difficult for water resource managers and policymakers to utilize (Degefu et al., 2018). Moreover, due to the lack of validation of most global hydrological models used to simulate runoff, there may be large biases in water supply assessment. A wealth

of previous studies in China have explored the general feature of water stress in the YRB at different spatial scales, ranging from provincial or prefectural (Zhao et al., 2015; Huang et al., 2023), to river basin scale (Yin et al., 2020), to sub-basin scale (Zhou et al., 2019; Sun et al., 2021; Xu et al., 2022), to grid scales (Zhuo et al., 2016; Liu et al., 2019). Recently, considering quality requirements, a comprehensive series of

assessments of nationwide water scarcity at multiple temporal and geographic scales has been performed in China (Ma et al., 2020a), which has markedly advanced our understanding of current water scarcity conditions. However, these assessments often covered only short periods, overlooked environmental water needs, or suffered from the same problems faced by global water stress assessments. In addition, upstream

inflows and water consumption were usually not taken into account. A neglect of upstream water availability means that downstream water scarcity will be



overestimated (Liu et al., 2019; Munia et al., 2020). More importantly, human water usage estimates in both the historical and future have mostly been based on macroscale socio-economic activity data (such as gross domestic product and population) (Wada et

al., 2016a; Yin et al., 2017); this may underestimate the effects of technological factors and water conservancy measures and bring uncertainties in water scarcity assessments (Zhou et al., 2020; Huang et al., 2021). A long-term water withdrawal dataset at the prefectural scale based on nationally coordinated surveys has recently been constructed by Zhou et al. (2020). The use of this newly developed dataset could bring new insights

into the changing pattern of historical water scarcity and result in more reliable future projections. A previous study has explored the spatiotemporal features of water scarcity in the YRB (Zhang et al., 2024). Nevertheless, the role of water withdrawal (i.e., different sectors) and water availability components in determining water scarcity remains unclear. Meanwhile, potentially feasible solutions in terms of water use

efficiency improvement for mitigating future water scarcity have not been quantified.

Here, we provide a spatially explicit assessment of water scarcity in the YRB at the sub-basin scale, taking into account environmental flow requirements (EFR) and upstream flows. The objectives of our study were to: (1) assess the evolution of water scarcity over the past five decades in terms of critical indicators, including the intensity,

frequency, duration, and exposure of population; (2) identify the dominant driver of changes in water stress during 1965–2013; and (3) quantify future water stress and explore potential solutions. Our findings provide valuable information for designing policies towards integrated water resource management in the YRB and other similar basins.


## 2 Materials and methods

Multiple datasets and methods were used to develop an integrated analytical framework, including multi-dimensional water stress indicators (i.e., intensity, duration, frequency, and exposed population), driving factors of changes in historical water stress, and water

deficits in the future along with potential solutions to address these (Fig. 1). A more





detailed description can be seen in the following section.

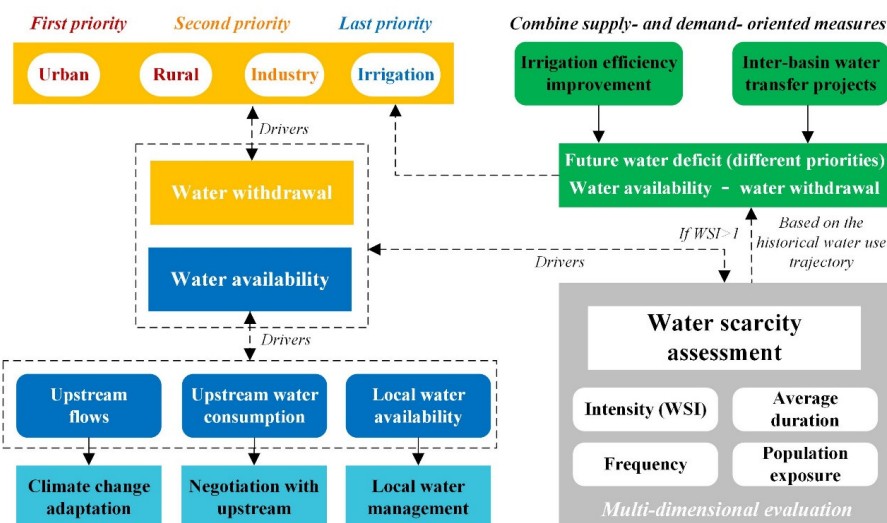

**Figure 1.** Framework for water scarcity assessment.

### 2.1. Study area

The YRB is the second-largest basin in China, with a total drainage area of $79.5 \times 10^4$ km$^2$ and a mainstream length of 5464 km. It runs through nine provinces or municipalities and three geomorphological units: the Qinghai–Tibet Plateau, Loess Plateau, and North China Plain (Fig. 2). Our study focused on the areas above the Huayuankou hydrological station, the outlet of the middle reaches of the YRB, owing to the negligible runoff downstream of this. Based on the Soil and Water Assessment Tool (SWAT), the study area was further divided into 425 sub-basins. Most parts of the YRB are arid or semi-arid regions, with the mean annual precipitation being 450 mm. The rapid expansion of irrigated agriculture has induced a very large irrigation water demand, while population growth and socio-economic development have led to an increase in domestic and industrial water use (Zhou et al., 2020), leading to an intensified water crisis in this water-limited basin.



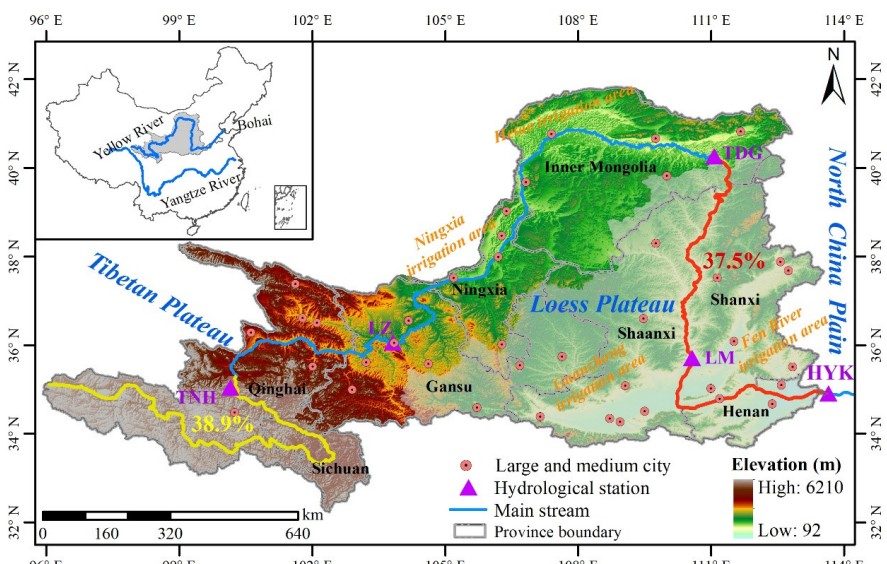

**Figure 2.** Location of the study area. TNH = Tangnaihai, LZ = Lanzhou, TDG =
Toudaoguai, LM = Longmen, and HYK = Huayuankou. Yellow and red numbers
indicate the ratio of mean annual natural runoff above the TNH station and the section
between TDG and HYK station to that of the HYK station, respectively.

### 2.2. Critical indicators of water scarcity

The water scarcity index (WSI), which is widely used to assess water stress intensity,
is defined as the ratio of water withdrawal to water availability:

$$\text{WSI}_{i,m} = \frac{WW_{i,m}}{\text{WA}_{i,m}} \ , \tag{1}$$

where, $WW_{i,m}$ is the total water withdrawal (irrigation, industry, urban, and rural) in
sub-basin $i$ and month $m$; $\text{WA}_{i,m}$ is the total water availability for sub-basin $i$ in month
$m$, which consists of the locally generated runoff and incoming discharge from upstream
sub-basins taking into account the EFR and upstream water consumption (Liu et al.,
2019):

$$\text{WA}_{i,m} = R_{i,m} + \sum_{j=0}^{i}(Q_{j,m} - WC_m) - EFR_{i,m} \ , \tag{2}$$





where, $R_{i,m}$ is the local water yield, including surface, baseflow, and lateral flow, simulated by the SWAT model; $Q_{j,m}$ is the discharge entering sub-basin $i$ from all upstream sub-basins $j$ in month $m$; $WC_m$ is the total upstream water consumption in month $m$; and $EFR_{i,m}$ is the environmental flow requirements in sub-basin $i$ and month $m$, which is determined based on the variable monthly flow method (Pastor et al., 2014).

In addition to the severity of water stress (WSI), frequency and average duration were also used to describe water scarcity (Veldkamp et al., 2017):

$$Frequency_i = \frac{NMWS_i}{TM} \ , \tag{3}$$

$$Average \ duration_i = \frac{NMWS_i}{NEWS_i} \ . \tag{4}$$

where, $NMWS_i$ is the number of months with WSI>1 in sub-basin $i$; TM is the total number of months in different periods (e.g., 10 years = 120 months); $NEWS_i$ is the number of water scarcity events in sub-basin $i$.

In terms of the population exposed to water scarcity, we categorized sub-basins into four groups with varying WSI values between two consecutive periods: moving into/out of water scarcity and alleviation/aggravation of water scarcity (Table S1).

## 2.3. Attribution analysis

To identify the dominant factor affecting the changing pattern of water stress at the sub-basin scale, we first calculated the differences in WSI between consecutive time steps (ΔWSI) at the decadal scale (P1: 1965–1979, P2: 1980–1989, P3: 1990–1999, and P4: 2000–2013).

$$\Delta WSI = WSI_{t+1} - WSI_t \ , \tag{5}$$

Then, the relative contributions of water withdrawal (different water use sectors) and water availability (climate change) to changes in WSI (ΔWSI) were obtained by keeping one factor constant, as follows:





$$\Delta \text{WSI}_{WW} = \left( \frac{WW_{t+1}}{WA_t} - \frac{WW_t}{WA_t} \right) / |\Delta \text{WSI}| \ , \tag{6}$$

$$\Delta \text{WSI}_{WA} = \left( \frac{WW_t}{WA_{t+1}} - \frac{WW_t}{WA_t} \right) / |\Delta \text{WSI}| \ , \tag{7}$$

$$Driver_{WSI} = \text{maximum} \left( |\Delta \text{WSI}_{WW}, \Delta \text{WSI}_{WA}| \right) \ . \tag{8}$$

where, $\Delta \text{WSI}_{WW}$ and $\Delta \text{WSI}_{WA}$ are the contributions of water withdrawal and water availability, respectively.

According to equation 2, water availability comprises local water availability (ΔLocal WA), upstream flows (ΔUp WA), and upstream water consumption (ΔUp WC). We further explored the relative changes in components of water availability for each sub-basin between two consecutive periods (Munia et al., 2020).

$$\Delta \text{Local WA} = \text{Local WA}_{t+1} - \text{Local WA}_t \ , \tag{9}$$

$$\Delta \text{Up WA} = \text{Up WA}_{t+1} - \text{Up WA}_t \ , \tag{10}$$

$$\Delta \text{Up WC} = \text{Up WC}_{t+1} - \text{Up WC}_t \ , \tag{11}$$

Similarly, the dominant driver of water availability ($Driver_{WA}$) was calculated as follows:

$$Driver_{WA} = \text{maximum} \left( |\Delta \text{Local WA}, \Delta \text{Up WA}, \Delta \text{Up WC}| \right) \ . \tag{12}$$

Besides, to assess the impact of vegetation restoration on water availability, we re-run the SWAT model with the fixed land use in 1990 and varied climatic conditions. The WSI and population exposed to water scarcity in P4 have been recalculated.

### 2.4. Water availability and water use data processing

Natural water availability data for the period 1965–2013 at the sub-basin scale are based on the SWAT model simulation that have been validated against natural discharges from hydrological stations. Historical annual prefectural-level human sectoral water withdrawal data, including irrigation (7 sub-sectors), industry (11), urban (2), and rural





(2) water uses, were obtained from Zhou et al. (2020), which is based on National Water

Resources Assessment Programs (1965–2000) and Water Resources Bulletins (2001–

2013). This dataset was first disaggregated at the 1-km spatial resolution grid scale

according to land use and population density data (https://www.resdc.cn/). Specifically,

irrigation water withdrawals were downscaled based on the irrigated cropland land use

and net irrigation requirement, as follows:

$$IRW_{j,m} = \frac{I_{j,m} \times A_j}{\sum_{m=1}^{12} \sum_{k=1}^{N} I_{k,m} \times A_k} \times IRW_a \qquad (13)$$

where, $IRW_{j,m}$ is the irrigation water withdrawal of grid cell $j$ in month $m$; $I_{j,m}$ is the

net irrigation requirement of grid cell $j$ in month $m$ for different crops, which was

calculated as the difference between the reference crop evapotranspiration and effective

precipitation; $A_j$ is the irrigated area; $IRW_a$ is the annual irrigation water withdrawal

in the city where grid cell $j$ is located; $N$ is the number of irrigated cropland grid cells

in that city.

Similarly, industrial water withdrawals were downscaled according to the maps of

industrial and mining land, and assumed equally distributed within a year. The urban

and rural water withdrawals were disaggregated based on the urban and rural residential

areas, population density, and a monthly factor:

$$URW_{j,m} = \frac{Pop_j}{\sum_{m=1}^{12} \sum_{k=1}^{N} M_{k,m} \times Pop_k} \times URW_a \times M_{j,m} \qquad (14)$$

where, $URW_{j,m}$ is the monthly urban and rural water withdrawals in grid cell $j$; $Pop_j$

is the population in grid cell $j$ and $URW_a$ is the annual urban and rural water

withdrawals in the city where grid cell $j$ is located; $M_{j,m}$ is a monthly factor

considering seasonal water use variations (Huang et al., 2018).

Then, the grid-level water withdrawal datasets were aggregated at the sub-basin scale.

The detailed calculation process of historical water availability and water withdrawal

can be found in Zhang et al. (2024). The future annual water demand from different

sectors at the sub-basin scale in the 2030s were estimated based on the trajectories of



water withdrawals during recent decade. Specifically, the linear regression was used to
detect the trend of individual sectoral water withdrawal from 2000 to 2013 (P4 period).
If the trend is significant ($p<0.05$), the future predicted water demand was obtained
based on the corresponding water withdrawal for the base year (mean value of 2009–
2013) and this trend (equation 15), otherwise the base year water withdrawal was
adopted as the water demand in the 2030s.

$$FWD = WW_{to} + trend \times (t - t_0) \tag{15}$$

where, $FWD$ is the future water demand for each year during 2030–2039; $WW_{to}$ is
the water withdrawal for the base year; $trend$ is the linear slope of the water
withdrawal during P4 period ($10^4$ m$^3$ per year). The mean annual value of the ten years
(2030–2039) is denoted as 2030s.

**2.5. Future water stress and potential solutions**

Given the high uncertainty in climate change projections (Greve et al., 2018), we only
focused here on the impact of changes in human water usage on water scarcity. Thus,
the future WSI was estimated based on the water demand in the 2030s (see section 2.4)
and water availability in P4. The future water deficit was further calculated by
subtracting water demand from water availability (fixed in P4) for each sub-basin with
WSI>1. We considered water use in different priority sectors. The first priority was
urban and rural water demand, second priority was industrial water demand, and last
priority water was irrigation. From first priority to last priority, a greater water stress
equated to greater socio-economic loss. Competition between the agricultural and other
sectors for water is increasing. Irrigation water usage accounts for the largest proportion
of total water demand, which also provides the most practical solution for alleviating
water scarcity across all sectors. We thus quantified the required improvements in
irrigation efficiency (IE) to address future water deficits under different priority
assumptions. Meanwhile, based on the predicted IE increase rate from Zhao et al. (2015)
(Table S2), we further calculated the contributions of IE improvement to the regional





water deficit.

## 3 Results

### 3.1. Multi-dimensional assessment of water scarcity and attribution of changes in
WSI

As shown in Fig. 3a, regional WSI consistently increased during the last five decades,
almost doubling from 0.59 in P1 to 1.17 in P4. This exacerbated water stress was
dominated by an increase in water withdrawals moving from one sub-period to the next,
except for the phase from P2 to P3 when water withdrawals and climate change
combined to an increased WSI. In other sub-periods, climate change had a minor
alleviation effect on water stress. Water stress hotspots were mainly found in the
Lanzhou–Toudaoguai section, characterized by high water demand but limited water
availability (Fig. S1). In contrast, water stress was low in regions above Lanzhou station
and the middle parts of the Loess Plateau, with annual WSI values < 0.2. The regional
average duration of water scarcity increased from 3.3 months in P1 to 7 months in P4,
and the frequency increased from 0.43 in P1 to 0.65 in P3, then dropped slightly to 0.63
in P4 (Fig. 3b). These indicators showed a similar spatial distribution pattern to that of
severity (WSI value). For example, in areas where large irrigated districts or cities are
located (e.g., Lanzhou–Toudaoguai section and eastern parts of Shanxi province), the
290 frequency was > 0.6 and average duration was at least four months, with some sub-
basins facing year-round water scarcity (Figs. S2a and S2b).

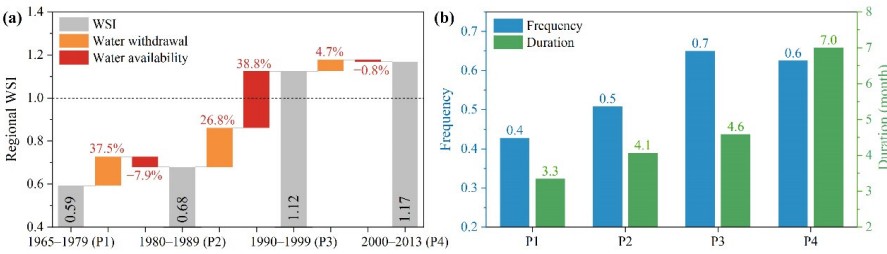

**Figure 3.** (a) Evolution of the regional water scarcity index (WSI) and its drivers. (b)
Frequency and average duration of water scarcity.

Generally, the change directions of WSI, frequency, and duration of water scarcity presented similar patterns (Figs. 4 and S2c–h). Specifically, most sub-basins experienced increases in WSI, frequency, and duration of water scarcity from P1 to P2, except for some southern parts of the YRB and the western Hetao irrigation area (Figs. 4a and 4d). From P2 to P3, decreases in the three indicators were concentrated in the eastern YRB, whereas, from P3 to P4, changes were not spatially confined. Overall, the WSI decreased in more sub-basins compared with the other indicators, and there were fewer areas where both frequency and duration decreased (Figs. 4c and 4f).

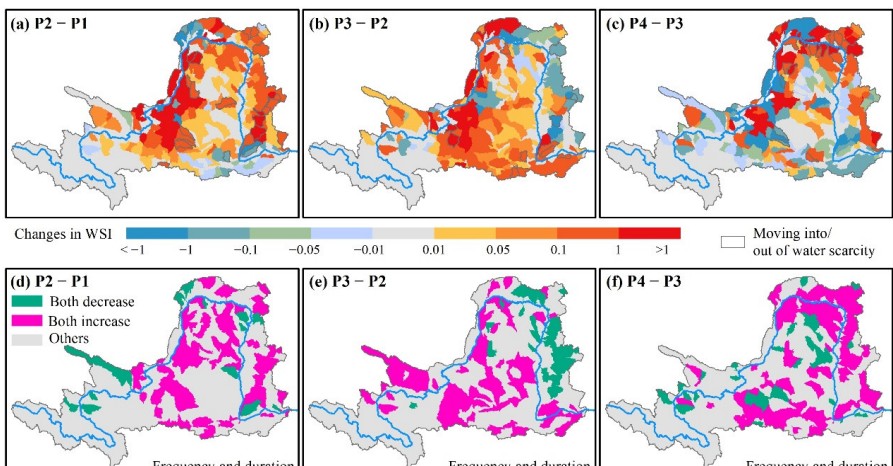

**Figure 4.** Changes in the (a, b, and c) water scarcity index (WSI), (d, e, and f) frequency, and duration of water scarcity.

Further analysis showed that the population moving out of and experiencing alleviation in water scarcity accounted for 0.2%–2.7% and 3.2%–13.9%, respectively, of the total population during different periods; this was always much lower than the proportion of the population moving into (4%–11.2%) or experiencing an aggravation (14.1%–17.7%) in water scarcity in the corresponding periods. Much greater differences were found in the proportion of the population experiencing both changes in frequency and duration of water scarcity; 33.1%–34.9% of the population experienced an increase in both indicators, while only 4.9%–10.1% experienced a



decrease in both indicators (Fig. 5). In summary, changes in critical indicators of water scarcity suggest that the YRB has been facing increasingly unfavorable water crisis during the last five decades.

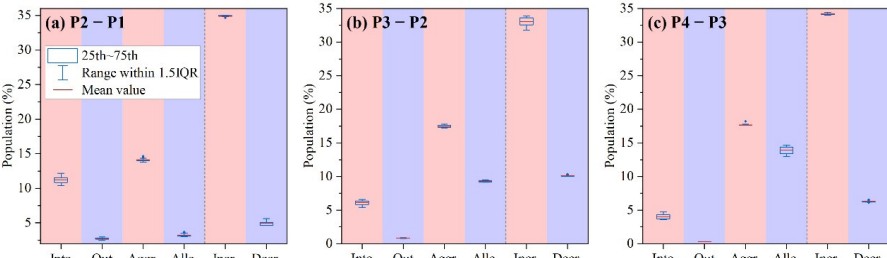

**Figure 5.** Proportion of the population moving into (Into)/out of (Out) water scarcity, experiencing aggravated (Aggr.)/alleviated (Alle.) water scarcity, and (right-hand side of dashed line) experiencing both increases (Incr.) and decreases (Decr.) in both frequency and average duration of water scarcity between two consecutive periods.

However, the dominant drivers of changes in WSI were not the same (Figs. 6 and S3). Specifically, the increased WSI in Gansu province can mainly be ascribed to an increase in wheat irrigation resulting from the expansion of the irrigated areas from P1 to P2, whereas in other regions it was related to decreased water availability owing to climate change (Figs. 6a, 6d, 6g, S3a, and S3d). Climate change also led to some alleviation of water stress in Shanxi province from P2 to P3 but exerted the opposite effect in central and southern parts of Shaanxi province (Fig. 6b). Irrigation of wheat, rice, and vegetables and fruit was the key factor leading to intensified water stress in the Lanzhou–Toudaoguai section (Figs. 6e, 6h–6k, S3b, and 3e). The recent two decades witnessed mixed and more spatially distributed patterns of changes in WSI, and irrigation was not the primary reason for increased WSI in most sub-basins (Figs. 6c and 6f). Meanwhile, we found that despite the rapid growth of irrigation in vegetables and fruit in southern Gansu and Shaanxi provinces since the 1980s, its influence on changes in WSI was less than that of water availability (Figs. 6b and 6c), highlighting the climatic controls on changes in water stress in these regions.



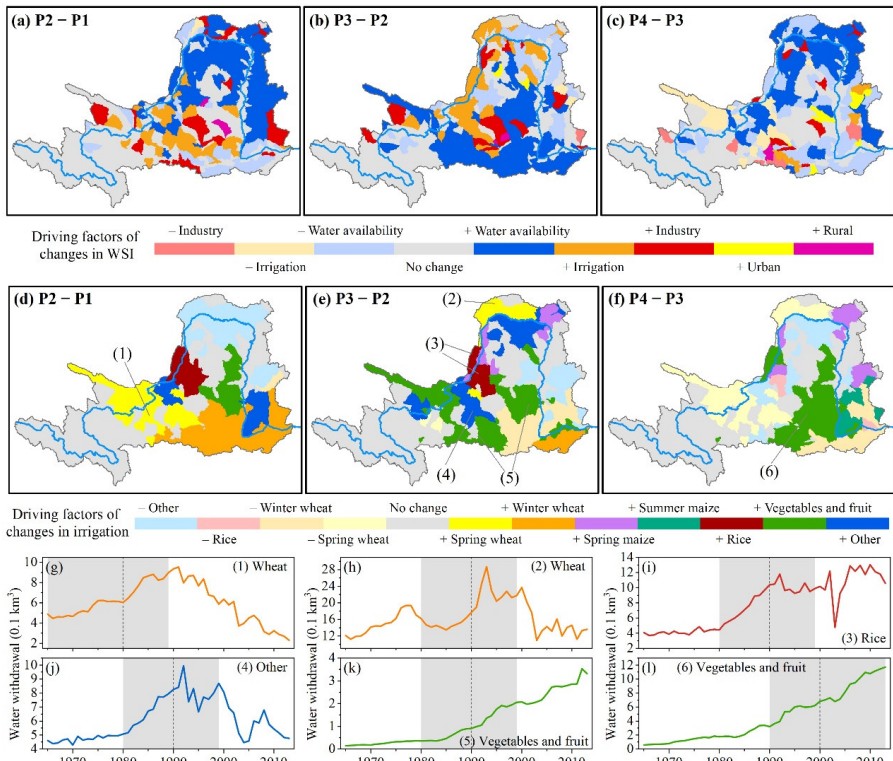

**Figure 6.** Driving factors of changes in (a, b, and c) WSI and (d, e, and f) irrigation between two consecutive periods at the sub-basin scale. For the drivers, a '+' prefix indicates a positive effect on ΔWSI or irrigation, whereas a '−' indicates a negative effect. (g-l) Interannual variation in water withdrawals from 1965 to 2013 in typical cities across the study area. Gray shadings show the periods: from P1 to P2 (g), from P2 to P3 (h–k), and from P3 to P4 (i). The dashed lines show the start year of the latter period (i.e., 1980 in g, 1990 in h–k, and 2000 in l). The numbers 1–6 refer to locations shown in panels d–f.

## 3.2. Important roles of local water availability and upstream flows

Given the important role of water availability in changes in water scarcity at the sub-basin scale, we further explored its drivers in terms of changes in local water availability, upstream flows, and upstream water consumption (see equations 9–12). We found that



upstream flows were responsible for changes in net water availability in 36%–40%

(152–176) of the sub-basins (Fig. 7a), most of which were located along the main stem

of the Yellow River (Fig. S4). The upper regions above Tangnaihai and Lanzhou

hydrological stations had the largest shares of total natural flows (Huayuankou station),

with mean annual ratios of 0.39 and 0.63, respectively. More importantly, these ratios

consistently increased during the study period (Fig. 7b), highlighting the increasingly

dominant role of these regions in determining the total water availability of the whole

basin owing to climate change. However, this is also mostly beyond the control of local

decision-makers.

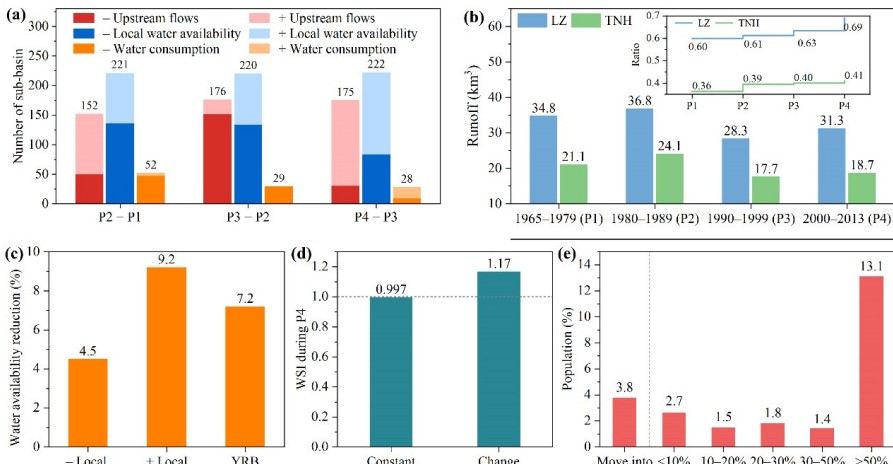

**Figure 7.** (a) Number of sub-basins experiencing different drivers of total water

availability between two consecutive periods. For the drivers, a '+' prefix indicates a

positive effect on water availability, whereas a '–' indicates a negative effect. (b) Natural

runoff at the Tangnaihai (TNH) and Lanzhou (LZ) stations during different periods. The

inset shows the runoff ratio of TNH and LZ to the Huayuankou station. (c) Water

availability reduction of local water availability-dominated sub-basins (− local and +

local) and the YRB with and without vegetation restoration (i.e., land use in 1990). (d)

WSI in P4 with (Change) and without (Constant) vegetation restoration. (e) Population

moving into water scarcity and experiencing aggravated water scarcity (increased WSI

expressed as a percentage) owing to vegetation restoration.



The effects of local water resource management were also prominent in half of the sub-basins (~220) (Fig. 7a). Owing to the implementation of large-scale ecological restoration projects (e.g., the Grain for Green Program) since the end of the 1990s, the vegetation coverage of the YRB has been significantly improved and large amounts of cropland and barren land were converted into forest and grassland (Figs. S5a–5c). However, the underlying surface changes reduced local water availability as a result of increased evapotranspiration (Fig. S5d). In terms of local water availability-dominant sub-basins, land cover transitions led to a mean annual reduction in natural flows of 4.5%/9.2% (Fig. 7c). Over the entire basin, this generally decreased natural flows by 7.2% during P4, giving a corresponding regional WSI of 0.997 (Fig. 7d), i.e., moving out of water scarcity or just approaching the threshold (WSI = 1). Vegetation restorations led to 3.8% of the regional population moving into water scarcity and a > 50% increase in WSI for 13.1% of the total population (Fig. 7e). This implies that local vegetation restoration in some places should be approached with caution to avoid water stress. For some sub-basins (approximately 10%) where water availability is controlled by changes in upstream water consumption, integrated water management strategies, such as water resource recycling, should be considered to indirectly reduce water consumption and meet the water usage needs of downstream areas (Zhou et al., 2021).

### 3.3. Future water stress and potential solutions

Total water usage in the 2030s is projected to decrease in northwestern parts of the YRB (except for Ningxia regions) but increase in southeastern parts (Fig. S6), mainly ascribed to reduction of irrigation (Fig. S7a). There is also a projected growth in industrial and urban water usage, with rates of increase generally above 20% and 100%, respectively (Figs. S7b and S7c). At the regional scale, despite the continuous increase in water use for maize, and vegetables and fruit (Fig. 8a), irrigation is projected to decrease by 10.7%. Urban water use shows the largest rate of increase (117%), followed by industrial (63.8%) and rural (1.4%) use. In total, the water use of the study area in the 2030s is projected to be 37.4 km$^3$ (12.6% higher than the current 33.2 km$^3$ during





the period 2000–2013), 55.5% (20.8 km³) of which will be contributed by irrigation

(Fig. 8b). The industrial sector will have the second-largest share of total water use

(24.2%), followed by urban (16%), then rural (4.3%).

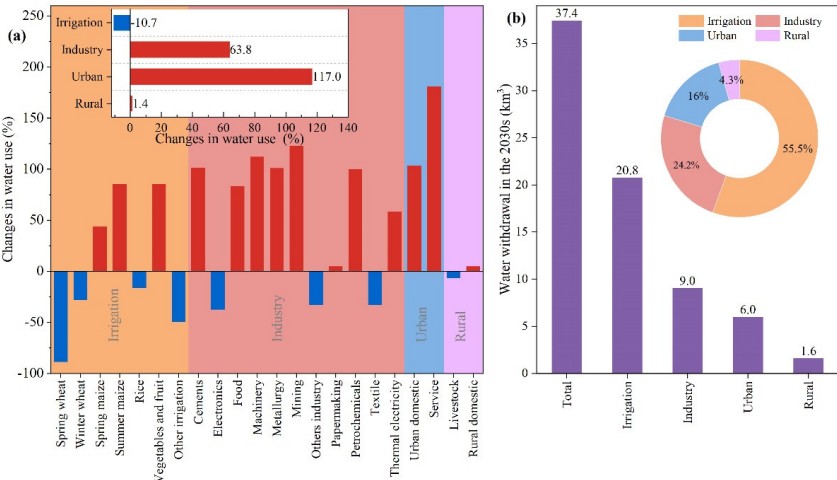

**Figure 8.** (a) Changes in water use of different sectors from the recent decade (2000–

2013) to the 2030s (%). (b) Water use in the 2030s and the proportion of use by different

sectors (pie chart).

Improving water use efficiency is a key solution to alleviate water scarcity. Thus, we

first determined how much water stress could be alleviated by improving water

efficiency by 10% via technical and economic progress in different sectors. We focused

on the top 5% of water use sub-basin sectors (Fig. S8), the combination of which

accounted for 43.8% of all water withdrawals in the 2030s (Fig. 9a). In other words, a

decrease in water use intensity of 10% in these sub-basin sectors could save 4.38% of

the total water use in the YRB. Four of the top five sectors were related to irrigation

(Fig. 9b), with the largest savings being vegetables and fruit (1.06%). In two other

scenarios, when an efficiency improvement of 10% in (1) only irrigation and (2) all

sectors was attained (Fig. 9c), slightly higher water savings were estimated (5.55% and

10%, respectively). Thus, these water savings could reduce the WSI to a certain extent

(from 1.25 to 1.13–1.2), but cannot entirely solve the water scarcity problem of the



YRB. Between the recent decade and the 2030s, in the absence of any water use

efficiency action, 23% (11.8%) and 5.7% (0.7%) of the population (2013) will

experience aggravated (alleviated) and be moved into (out of) water scarcity conditions,

respectively. An additional 2.8%–4.6% of the total population would be benefit from

water use efficiency improvements (Fig. 9d).

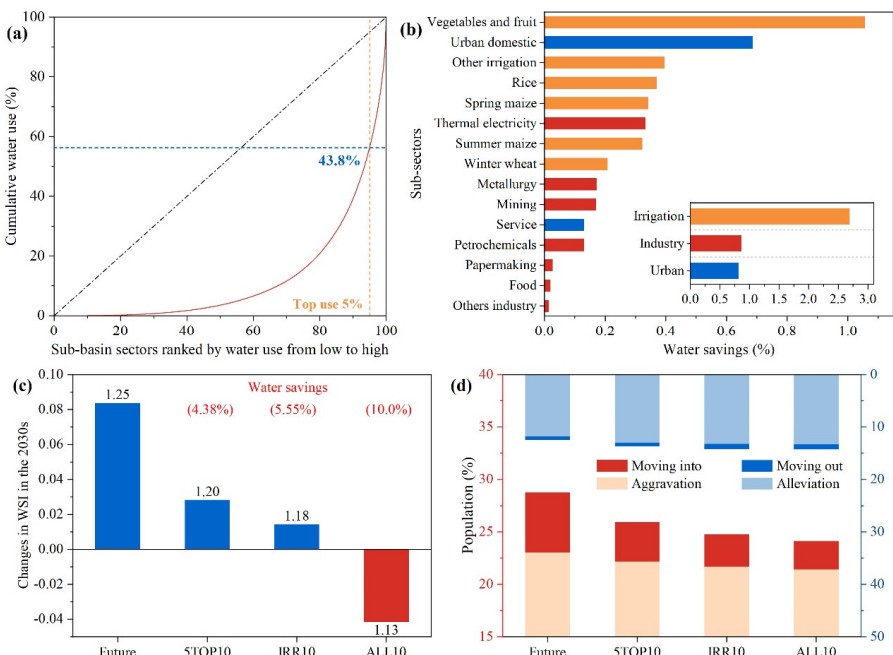

**Figure 9.** (a) Water-withdrawal Lorenz curve depicted as sub-basin sector combinations

during the 2030s (in order of water withdrawal from low to high). (b) Water savings

from the top 5% water use sub-basin sector combinations (sub-sectors) with water use

efficiency increasing by 10%. The inset shows water savings from the major sectors. (c)

Changes in WSI from the recent decade (2000–2013) to the 2030s. 5TOP10, IRR10,

and ALL10 denote water use efficiency increasing by 10% for the top 5% water use

sub-basin sectors, only irrigated sectors, and all sectors, respectively. Black numbers on

top of the bars are WSI values in the 2030s. Red numbers in bracket are the water

savings for the above three cases. (d) Percentage of the population experiencing

aggravated/alleviated water scarcity and moving into/out of water scarcity.



Irrigation has the largest share of total water use in the future, thus providing the most feasible water stress mitigation among all sectors (Figs. 8 and 9). It was found that, if urban, rural, and irrigation water usage were considered (first and last priority, see Section 2.5), 11 sub-basins (9% of the total sub-basins experiencing water scarcity) would not be under water scarcity conditions in the absence of additional measures. Improving irrigation efficiency (IE) by 30% can address water deficit in 15 sub-basins whereas further IE increases above 50% would be necessary in 36 sub-basins (Fig. 10a). For the other 53 sub-basins facing water scarcity, improvements in irrigation efficiency will not be plausible, given that their water deficits already exceed the irrigation water demand. Thus, other measures, such as water savings from other sectors or additional water resources from inter-basin water transfer projects or wastewater reuse should be advocated. When all sectoral water usages need to be fulfilled (Fig. 10b), 66% of the total sub-basins (79) that face water scarcity in the 2030s cannot be escaped by raising IE alone. In contrast, 17 sub-basins could overcome their water deficits with IE improvement up to 50%. Further analysis showed that the water deficit of the YRB in the 2030s would be 0.87–10 $km^3$ when considering sectoral water use with different priorities, respectively (Fig. 10c). The possible improvement of irrigation efficiency in the future (Table S2) could solve 26% of the maximum water deficit (all sector demands are met), leading to a net water deficit of 7.39 $km^3$. From the perspective of water supply, inter-basin water diversion project is regarded as important measure to alleviate the severe water stress problems experienced in northern China. The Hanjiang-to-Weihe River (HWR) project and the South-to-North Water Diversion (SNWD) project are the two most important projects for the YRB. The total water coming from the HWR project to Guanzhong Plain and from the Middle Route of the SNWD project to Henan province (SN-HN) would be 5.27 $km^3$ (Fig. 10d). The planning Western Route of the SNWD project (SN-WR), which links the headwaters of the Yangtze and YRB, is 8 $km^3$ in the first phase. Combined these projects with IE improvements, the water scarcity in the YRB can be effectively alleviated.



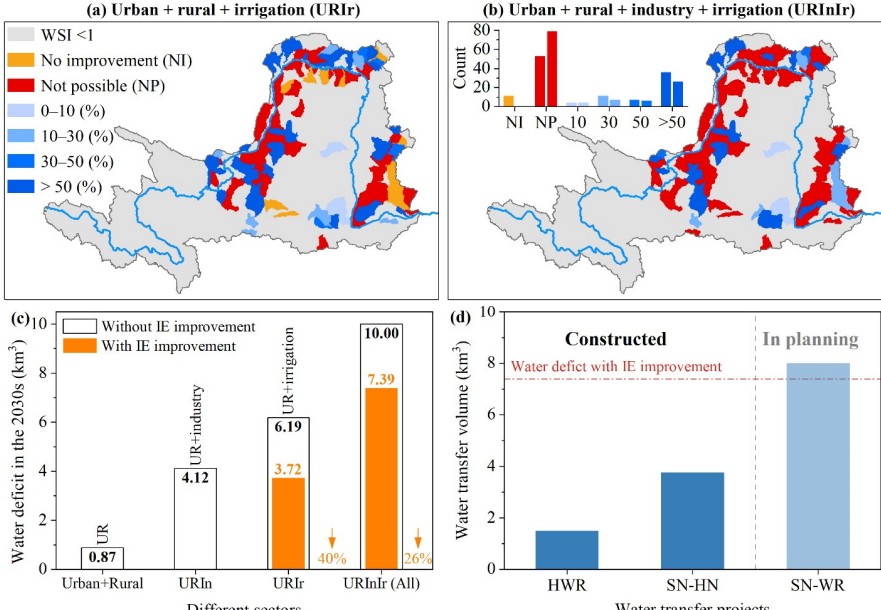

**Figure 10.** Sub-basins where an improvement in irrigation efficiency (IE) could help to solve water scarcity in the 2030s: (a) urban, rural, and irrigation (URIr) and (b) all sectors including urban, rural, industry, and irrigation (URInIr). Red color denotes sub-basins where improvement in IE > 100% or water deficit would exceed the irrigation demand. The inset in panel b shows the number of sub-basins with different IE improvement levels with the consideration of URIr (left bar) and URInIr (right bar). (c) Water deficit in the 2030s under different sectoral water demand. (d) Water transfer volumes to the study area from the HWR project, SNWD project to Henan Province (SN-HN), and Western Route of the SNWD project (SN-WR) in the first phase.

## 4 Discussion

### 4.1. Water stress in the past and futures: comparison with other studies

Water stress hotspots were mainly concentrated in areas between the Lanzhou and Toudaoguai hydrological stations and in large cities with higher water demands (Fig. S1); this was similar to the findings of previous studies (Ma et al., 2020a; Xie et al., 2020; Sun et al., 2021). However, the severity of water stress was not the same as that



reported in previous studies because of the choice of different water withdrawals, water availability datasets, and proportion of EFR in the WSI estimation. We also found that water stress over the whole study area has generally intensified during the past five decades, which also generally corroborates previous findings at different scales (Liu et al., 2019; Zhou et al., 2019; Huang et al., 2021; Huang et al., 2023). In terms of the

driving factors of changes in water stress, previous studies have concluded that a rapid growth in water use was the primary factor leading to an increased WSI in northern China during the past four decades (Wada et al., 2011; Huang et al., 2021), while increased water availability owing to enhanced precipitation was the main contributor to alleviating water scarcity in the YRB during 2001–2020 (Huang et al., 2023). These

results were confirmed by our study. By using survey-based water withdrawal datasets and SWAT simulations, we found that changes in water stress and its drivers in the YRB were sub-basin and study period dependent (Fig. 4). In addition, our study further isolated the contribution of water withdrawals by various water use sectors on the evolution of water stress at the sub-basin scale (Fig. 6); this was rarely considered in

previous studies. We found that irrigation water use, for wheat, rice, and vegetables and fruit, dominated increases in the WSI in northwestern parts of the basin (except for the source regions) before 2000. In some sub-basins, industry was the primary driver of WSI increases. However, owing to the implementation of a series of laws, regulations, action plans, and technology related to saving water promoted by China's State Council

and Ministries after 2000 (Zhao et al., 2015; Zhou et al., 2020), a widespread decrease in human water withdrawal in the northwestern parts of the basin and thus a regional deceleration in the rate of WSI increase was observed (Figs. 3a and S3c). Despite some increase in water withdrawals in the middle and southern parts of the basin owing to the irrigation of vegetables and fruit, the withdrawal magnitudes were lower than those

of water availability (Figs. 6 and S3); hence water stress conditions were alleviated.

Because the decrease in irrigation water use was offset by an increase in water use from the other three sectors (urban, rural, and industry), the final estimated total water use in the 2030s was calculated as 37.4 km$^3$ based on the trajectory of historical water





withdrawals (Fig. 8). When further water withdrawals of ~12 km$^3$ below the

Huayuankou station were taken into account, the total water demand of the YRB was

approximately 50 km$^3$. This value is 5–15 km$^3$ lower than that reported by Yin et al.

(2020), based on different global hydrological models (GHM) and shared socio-

economic pathways (SSP) combinations. One reason for this is that large-scale GHMs,

used to simulate sectoral water withdrawal, assume simple changes in technological

factors and the effect of water conservancy measures was often negligible (Haddeland

et al., 2014; Liu et al., 2019). When water availability remained stable (constant at the

level of 2000–2013), the water stress was projected to be further aggravated, especially

in the southern parts of the basin (Fig. S6), which was also reported by earlier studies

(Yin et al., 2017; Liu et al., 2019; Yin et al., 2020; Yang et al., 2023).


**4.2. Broader implications for water resources management**

In terms of water supply, the upper YRB including the source regions (above the

Tangnaihai station) provides a vital water source for the whole YRB. However, water

availability in this region is highly sensitive to global warming (Zhang et al., 2014;

Kuang and Jiao, 2016; Ji et al., 2023), which can then further exert a considerable

impact on the security of food, energy, and water in downstream regions (Cui et al.,

2023). Meltwater from snow and glaciers can help mitigate water stress for both local

and downstream areas. However, meltwater acting as an additional source of water is

not sustainable in the long term (Wang et al., 2023). Influenced by global warming, the

terrestrial water storage deficit is predicted to expand northwards on the Tibetan Plateau

by the end of the century, which threatens the sustainability of water supplies in the

upper YRB (Zhang et al., 2023b). Worse still, the acceleration of glacier mass loss may

bring about a series of unintended and detrimental environmental and ecological

consequences (Hugonnet et al., 2021). Therefore, it is necessary to urgently reduce

carbon emissions and slow climate warming to alleviate the water stress of millions of

people living downstream in the YRB.

Meanwhile, we found that total water availability in more than half of the sub-basins was controlled by local water yields (Fig. 7a), underscoring the importance of local water management strategies. Owing to several ambitious programs to conserve and

expand forests (Bryan et al., 2018), vegetation greening of the YRB has been strikingly prominent during recent years (Zhang et al., 2023a); however, a strain on water resources has been observed in our study (Figs. 7c–e and S5) and many other previous studies (Feng et al., 2016; Wang et al., 2019; Zhang et al., 2023c; Yao et al., 2024). By fixing the land cover in 1990 and varying climatic conditions, we estimated that

vegetation restoration led to a reduction in runoff of 7.2% over the YRB during the recent decade. This is generally in line with results from semi-arid and arid regions in China (Zhang et al., 2018), where vegetation restoration resulted in 8.5% and 11.7% reductions in runoff in the 1990s and 2000s, respectively. In those areas already facing a decrease in water availability induced by climate change, vegetation greening

undoubtedly exacerbated the water crisis. Therefore, to alleviate regional water stress, in addition to climate change adaptation strategies and substantial investment in hard-path infrastructure, nature-based solutions can also be adopted; for example, controlling the scale of reforestation and conducting appropriate grazing activities in excessively restored grassland (Liang et al., 2019; Deng et al., 2023). This can potentially achieve

a triple-win in economic development, ecological protection, and water resource security.

Despite considerable changes in economic structure, irrigation still accounts for the largest share of total water use in the 2030s; therefore, its efficiency improvement likely provides the most feasible solution for mitigating water stress in the YRB. Industrial

water use also comprised a large share of total water use, but most efforts in promoting efficiency gains in the industrial sector are currently in the pilot projects requiring high investment for low return (Blanke et al., 2007; Zhao et al., 2015). The YRB is a water-scarce area, where it is difficult to entirely solve the water shortage problem by focusing only on the supply or demand perspective. As shown in Fig. 10b, we found that it will

be almost impossible to meet both productive and domestic water demands through


irrigation efficiency improvements in most sub-basins in the 2030s. Relying on external inter-basin water transfer projects can mitigate the water scarcity in the region, but these projects often have substantial ecological and social impacts (Webber et al., 2021; Liu et al., 2023). Irrigation efficiency improvements can free-up water resources and reduce

26% of the water deficit to meet all sectoral water demands (Fig. 10c), greatly alleviating the pressure on water diversion projects (Fig. 10d). More importantly, compared with large-scale infrastructure, these measures have lower environmental costs. In addition, the strict Three Red Lines in 2012 and recent implementation of the Yellow River Protection Law (2023) have provided legislative support for the

protection of water resources in the YRB, which may also alleviate the pressure on water resources to a certain extent. In summary, it will be necessary to combine various measures to solve the YRB water crisis.

### 4.3. Uncertainty

Our study contains several uncertainties. First, the impact of reservoirs was not considered in our study due to data inaccessibility at a finer time scale. Although of little impact on the decadal scale, this may result in overestimations of the frequency and duration of water scarcity. This is because there are many reservoirs in the YRB and occasionally monthly water scarcity in some sub-basins may be addressed or

alleviated through seasonal flow regulations (Ma et al., 2020a; Ji et al., 2023). Second, we may have overestimated the number of people exposed to water scarcity. In the real world, non-agricultural water use has a higher priority than agricultural water given the fact that limited water resources are firstly allocated to maintain human domestic demand (Flörke et al., 2018). Cropland is also generally far from urban areas, so water

scarcity in a sub-basin does not necessarily mean a shortage of domestic water. Finally, it is challenging to predict water withdrawals because future water demand will depend on the integrated effects of many factors related to economics, policies, and population (Bijl et al., 2018; Ren et al., 2024). We assumed that the water use trends of the historical period would continue in the future period. According to the predictions under various



SSPs, the population of the YRB and northern China will peak around 2030 (Yin et al.,
2017; Yin et al., 2020). However, the National Bureau of Statistics' latest demographic
assessments show that China's population has begun to show negative growth since
2022. This is bound to affect future economic growth and changes in water use.
Moreover, future projected water use partly incorporated the effect of water efficiency

improvements and current water-saving technologies have already been widely adopted
(Zhou et al., 2020). When water use efficiency increases, the marginal cost for
maintaining a decreasing water use intensity trend increases (Sun, 2023). Therefore, the
ability to conserve water would gradually diminish in the future and the sustained relief
of water stress through water efficiency improvement would be limited, highlighting

the important role of climate change in affecting water stress under increasingly
stringent water management strategies.

## 5 Conclusions

This study presents an integrated analytical framework to reveal a comprehensive

picture of a given water crisis in the YRB, including multiple water scarcity indicators,
driving factors of changes in WSI, and future predictions of water stress along with
potential feasible solutions. Generally speaking, analysis of critical indicators (WSI,
frequency, duration, and amount of people exposed to water scarcity) shows that the
water supply and demand situation in the YRB has evolved in an unfavorable direction

during the past five decades. Compared with the period before the 1980s, the regional
WSI, frequency, and duration of water scarcity had increased by 98.3%, 112%, and
46.5%, respectively, by the 2000s. Changes in the WSI and its drivers were sub-basins
and study periods dependent. Specifically, irrigation, including wheat, rice, and
vegetables and fruit, dominated increases in the WSI in northwestern parts of the basin

before 2000. In contrast, climate change was responsible for the changes in WSI in most
of the sub-basins during the most recent decade studied. Meanwhile, further analysis
showed that upstream flows and local water yields were the main factors contributing
to changes in total water availability at the sub-basin scale. Pressure from upstream





regions of the YRB on downstream water use has gradually increased and will be unsustainable in the future. Vegetation restoration led to a reduction in natural flow of 7.2% after 2000, aggravating regional water shortages.

On the basis of the historical water use trajectory, we predict that the total water demand of the YRB in the 2030s will be 37.4 km$^3$, largely resulting from increased urban and industrial use, partly offset by decreased irrigation. In this context, the water 635 crisis will worsen for 28.7% of the total population and ease for 12.5% of the population in the absence of any other action. Considering the sectoral water use with different priorities, the water deficit of the study area is predicted to be 0.87–10 km$^3$ in the 2030s. The possible improvement of irrigation efficiency could solve 26% of the water deficit (all sector demands are met), leading to a net water deficit of 7.39 km$^3$. These measures 640 could greatly alleviate the water pressure on external inter-basin water transfer projects, and have the benefit of lower economic and environmental costs. In summary, it is necessary to combine water supply- and demand-oriented measures to solve the water crisis in the YRB. The results of this study have important implications for coping with water scarcity not only in the YRB but also in other basins facing similar situations.


**Data availability**

Data will be made available on request (baoqzhang@lzu.edu.cn).

**Author contributions**

*Weibin Zhang*: Conceptualization, Data curation, Formal analysis, Methodology, 650 Validation, Visualization, Writing - original draft. *Xining Zhao*: Conceptualization, Funding acquisition, Validation, Resources, Writing - Review. *Xuerui Gao*: Data curation, Methodology, Writing - Review. *Wei Liang*: Conceptualization, Methodology, Validation. *Junyi Li*: Formal analysis, Resources, Writing - Review. *Baoqing Zhang*: Conceptualization, Formal analysis, Funding acquisition, Writing - Review and Editing.



**Competing interests**

The authors declare that they have no known competing financial interests or personal relationships that could have appeared to influence the work reported in this paper.

**Acknowledgments**

This research was jointly supported by the National Natural Science Foundation of
China (NSFC) (42125705 and 42041004), and National Key Research and Development Program of China (2021YFD1900700), The W. Liang would like to acknowledge support from the NSFC (42071144).

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
