# Peer review of "Spatially explicit assessment of water stress and potential mitigating solutions in a large"

_Hydrology and Earth System Sciences, 2024_

## Author Comment (AC1)

This study addresses the critical issue of water scarcity in the Yellow River Basin, presenting a novel and comprehensive assessment framework for analyzing water shortage. By utilizing a combination of models and statistical data, it explores the spatiotemporal changes in this severely water-deficient river basin over a span of nearly sixty years and provides insightful predictions for the future. The research question, which is a unique and crucial aspect in the field, is clearly defined, the thought process is clear, and the logical chain is complete, resulting in scientifically valuable information and conclusions. The figures are also well-designed. However, before formal publication, I have some suggestions for consideration by the authors:

Response: Thank you for your positive comments and valuable suggestions to improve the quality of our manuscript. We will make extensive modifications to our manuscript and data to make our results convincing. The detailed point-by-point responses are listed below.

**R1C1:** The abstract and the primary texts are too extended; it's recommended that they be simplified and the main contributions highlighted.

Response: Thank you for this suggestion. We will simplify the abstract and main body text to highlight the main findings in the revised version. For example, we will reduce the general implications of the results and some descriptions about future water use. Additionally, we will place certain figures, such as the driving factors of changes in irrigation, in the Supporting Information.

**R1C2:** In lines L103-L105, explicitly addressing the deficiencies in previous studies would be beneficial. For instance, what specific challenges do global water stress assessments face? This requires further clarification. Additionally, the decision not to consider upstream inflow and consumption, while not a significant issue in my view, should be explained or referenced to strengthen the paper's argument.

Response: Revised as follows:

Recently, considering quality requirements, a comprehensive series of assessments of

nationwide water scarcity at multiple temporal and geographic scales has been performed in China (Ma et al., 2020a). This has markedly advanced our understanding of current water scarcity conditions. **However, upstream inflows or water consumption were usually not taken into account in most of these assessments. The neglection of upstream water availability means that downstream water stress will be overestimated (Munia et al., 2020; Sun et al., 2021).** Previous work in China showed that the difference in population affected by severe water stress was 60% with and without consideration of upstream water resources, which is even larger in northern water-limited areas (Liu et al., 20119). **Incorporating upstream flows and water consumption offers a more reasonable assessment of water stress in the real world.** Some studies have made significant progress in understanding water stress in the YRB by considering upstream components, reservoir operations, or water transfer projects (Albers et al., 2021; Omer et al., 2020; Xie et al., 2020; Sun et al., 2021). **Yet, they often covered short periods (less than 20 years), thus precluding a comprehensive documentation of the temporal dynamics of water stress.**

References:

Albers, L. T., Schyns, J. F., Booij, M. J., and Zhuo, L.: Blue water footprint caps per sub-catchment to mitigate water scarcity in a large river basin: The case of the Yellow River in China, J. Hydrol., 603, 126992, https://doi.org/10.1016/j.jhydrol.2021.126992, 2021.

Liu, X., Tang, Q., Liu, W., Veldkamp, T. I. E., Boulange, J., Liu, J., Wada, Y., Huang, Z., and Yang, H.: A spatially explicit assessment of growing water stress in China from the past to the future, Earth's Future, 7, 1027-1043, https://doi.org/10.1029/2019EF001181, 2019.

Omer, A., Elagib, N. A., Zhuguo, M., Saleem, F., and Mohammed, A.: Water scarcity in the Yellow River Basin under future climate change and human activities, Sci. Total Environ., 749, 141446, https://doi.org/10.1016/j.scitotenv.2020.141446, 2020.

Sun, S., Zhou, X., Liu, H., Jiang, Y., Zhou, H., Zhang, C., and Fu, G.: Unraveling the

effect of inter-basin water transfer on reducing water scarcity and its inequality in China, Water Res., 194, 116931, https://doi.org/10.1016/j.watres.2021.116931, 2021.

Xie, P., Zhuo, L., Yang, X., Huang, H., Gao, X., and Wu, P.: Spatial-temporal variations in blue and green water resources, water footprints and water scarcities in a large river basin: A case for the Yellow River basin, J. Hydrol., 590, 125222, https://doi.org/10.1016/j.jhydrol.2020.125222, 2020.

**R1C3:** In line L122 regarding environmental flow requirements, there has been extensive discussion of this flow rate of Yellow River from different perspectives, leading to varied estimates; you need to elaborate on how you evaluated these considerations here.

Response: Yes, we agree with you. As you stated, there are indeed many methods to calculate environmental flow requirements (EFR), such as the Tennant method (Tennant, 1976), the Smakhtin method (Smakhtin et al., 2004), the presumptive standard method (Richter et al., 2012), and the variable monthly flow (VMF) method (Pastor et al., 2014). Liu et al. (2021) reported that the impacts of different EFRs on water stress assessment are substantial in many regions but comparatively minor in areas with intensive human water use, such as Northern China and India. Pastor et al. (2014) compared and tested different calculation methods for the estimation of EFR. **They showed that the VMF method was most compatible with actual environmental water requirements,** distinguishing between low-flow (60% of a water resource allocated to EFR) and high-flow conditions (30%). **Given its performance in the seasonal assessment of water availability (Veldkamp et al., 2017), we therefore adopted this EFR method in our study.** We acknowledge that our results will depend on the chosen method to determine EFR and thus water availability. **However, finding an appropriate EFR method that determines water stress is out of the scope of this study. In the revised version, we will add the reasons why we selected the VMF method and the specific algorithm.**

References:

Liu, X., Liu, W., Liu, L., Tang, Q., Liu, J., and Yang, H.: Environmental flow requirements largely reshape global surface water scarcity assessment, Environmental Research Letters, 16, 104029, 10.1088/1748-9326/ac27cb, 2021.

Pastor, A. V., Ludwig, F., Biemans, H., Hoff, H., and Kabat, P.: Accounting for environmental flow requirements in global water assessments, Hydrol. Earth Syst. Sci., 18, 5041-5059, 10.5194/hess-18-5041-2014, 2014.

Richter, B. D., Davis, M. M., Apse, C., and Konrad, C.: A presumptive standard for environmental flow protection, River Res. Appl., 28, 1312-1321, https://doi.org/10.1002/rra.1511, 2012.

Smakhtin, V., Revenga, C., and Döll, P.: A pilot global assessment of environmental water requirements and scarcity, Water International, 29, 307-317, 10.1080/02508060408691785, 2004.

Tennant, D. L.: Instream Flow Regimens for Fish, Wildlife, Recreation and Related Environmental Resources, Fisheries, 1, 6-10, https://doi.org/10.1577/1548-8446 (1976).

Veldkamp, T. I. E., Wada, Y., Aerts, J. C. J. H., Döll, P., Gosling, S. N., Liu, J., Masaki, Y., Oki, T., Ostberg, S., Pokhrel, Y., Satoh, Y., Kim, H., and Ward, P. J.: Water scarcity hotspots travel downstream due to human interventions in the 20th and 21st century, Nat. Commun., 8, 15697, 10.1038/ncomms15697, 2017.

**R1C4:** In Figure 2 and the introduction of the study area - clarify how you distinguish between upstream and downstream regions. Or rather, readers expect an understanding of how upstream usage creates pressure on downstream resources within your study area description.

Response: The sub-basin delineation is based on the Digital Elevation Model (DEM) using the SWAT model. Additionally, we used the "Burn in" tool in the SWAT model to ensure that the generated sub-basin reaches align with known stream locations. We will add the spatial distributions of river systems and sub-basins in the Supporting Information.

**R1C5:** On the Yellow River, policy-making & unified dispatching already dominate human water use. In your proposed analysis framework emphasizing coordination with upstream, does such coordination significantly impact the basin system?

Response: Yes, you are right. In the Yellow River basin, policy-making and unified dispatching are critical in managing human water use. These measures ensure a coordinated approach to water allocation and use, which is essential given the river's limited and highly variable water resources. In this study, we found that upstream flows were responsible for changes in net water availability in 36%–40% of the sub-basins. In these regions, effective coordination with upstream areas can reduce the frequency and severity of water scarcity and ensure a more reliable water supply for various users. Such coordination can greatly influence the basin system by enhancing water availability, alleviating stress, and bolstering overall resilience to climate change.

**R1C6:** We understand that Zhou's data only went up until 2013. Still, the authors need to find ways to explain that the lack of data from the recent decade will not lead to outdated trends affecting analysis or causing bias in conclusion.

Response: As you have noted, we agree that the lack of data from the recent decade is a potential limitation of the study. In the revised version, we will try our best to collect recent data on human water withdrawals (post-2014) and meteorological data to rerun the model and conduct a similar analysis to the current study. Additionally, to accurately assess the impacts of irrigation efficiency improvements on alleviating future water stress, newly published water management policies will also be incorporated. We expect that these efforts will make the manuscript more reliable. Please also refer to our response to Reviewer 2's comments (Comments 6 and 7).

**R1C7:** During the discussion, it is necessary to include a comparative analysis with previous calculations using other water shortage indicators. This will highlight how the contributions of this assessment can compensate for the shortcomings of previous ones.

Response: We sincerely appreciate the valuable comments. In the current manuscript, we have already compared our findings with other similar studies in the Yellow River

basin regarding water stress hotspots regions, future water demand, and drivers of changes in water stress (please see Section 4.1). **As you suggested, we will include more case studies in this basin to further strengthen our contributions.**

---

## Author Comment (AC2)

The manuscript provides an in-depth analysis of water scarcity in the Yellow River Basin (YRB) using an integrated analytical framework. The study spans from 1965 to 2013, focusing on critical indicators like the Water Scarcity Index (WSI), frequency, duration, and exposed population. It also projects future water demand and evaluates potential solutions, particularly improving irrigation efficiency.

General Comments

The study addresses a crucial topic in hydrology and water resources management, particularly for a region as significant as the YRB. Integrating historical data, model simulations, and future projections provides a comprehensive overview. However, several areas could benefit from further clarification and refinement.

Response: We greatly appreciate your professional review of our article. As you have noted, several issues need to be addressed. According to your valuable suggestions, we will make the necessary corrections to our current manuscript. The detailed corrections are listed below.

**R2C1:** In the introduction, the authors list the limitations of previous studies regarding water use and water withdrawal estimation, highlighting their implications for water stress assessments. While the study acknowledges these limitations and aims to address them, many of the same uncertainties are reiterated in the uncertainty section (section 4.3). This raises the question: if these limitations remain largely unresolved, why emphasize them in the introduction?

Response: Thanks for your comment. One of the key innovations of this study, which distinguishes it from previous research, is the use of **long-term historical observed water use data** (Zhou et al., 2020) to evaluate the evolution of water stress from 1965 to 2013. In response to your suggestion (Comment 7), we will make every effort to extend this dataset by collecting recent publicly available data (from 2014 to the present) and update the related analysis. Further**, we will predict future water demand based on the trajectories of updated observed water withdrawals during recent years as the business-as-usual scenario (please also see our responses to Comments 6 and**

**7)**. As stated in the uncertainty section, we acknowledge that this prediction may introduce some uncertainties and limitations due to technological advancements and population growth. However, predicting near-term future (2030s) water demand based on recent observed data may significantly reduce uncertainty. **More importantly, compared with methods based on macroscale socio-economic datasets, the use of observed water withdrawal data for water stress estimation ensures that the analysis of historical periods (spanning nearly six decades) is more reliable. Additionally, this water withdrawal dataset encompasses four major sectors, allowing us to further separate the effects of individual sectoral water withdrawals on changes in water stress.**

References:

Zhou, F., Bo, Y., Ciais, P., Dumas, P., Tang, Q., Wang, X., Liu, J., Zheng, C., Polcher, J., Yin, Z., Guimberteau, M., Peng, S., Ottle, C., Zhao, X., Zhao, J., Tan, Q., Chen, L., Shen, H., Yang, H., Piao, S., Wang, H., and Wada, Y.: Deceleration of China's human water use and its key drivers, Proc. Natl. Acad. Sci. USA, 117, 7702, 10.1073/pnas.1909902117, 2020.

**R2C2:** The authors' statements in the introduction about the limitations of previous studies using coarse spatial resolution global water scarcity assessments (e.g., 0.5° × 0.5° level) and neglecting upstream water availability are partly valid. However, there are existing studies that have addressed water scarcity in the Yellow River Basin (YRB) at a higher resolution, considering sub-basin scales and upstream water availability (e.g., Albers et al., 2021; Omer et al., 2020; Xie et al., 2020). Given this and the previous comment, the authors should revise the motivation section of the introduction accordingly.

Response: Thank you for providing these references. We will revise this section, as follows:

A substantial body of previous studies in China has explored the general features of water stress in the YRB at various spatial scales, ranging from provincial or prefectural

levels (Zhao et al., 2015; Huang et al., 2023), to river basin scale (Yin et al., 2020), sub-basin scale (Zhou et al., 2019; Xu et al., 2022), and grid scales (Zhuo et al., 2016; Liu et al., 2019). Recently, considering quality requirements, a comprehensive assessment of nationwide water stress at multiple temporal and geographic scales has been conducted in China (Ma et al., 2020a). These assessments have significantly advanced our understanding of current water scarcity conditions. However, upstream inflows and water consumption were usually not taken into account in these studies. The neglection of upstream water availability means that downstream water stress will be overestimated (Munia et al., 2020; Sun et al., 2021). Previous work in China showed that the difference in the population affected by severe water stress was 60% with and without consideration of upstream water resources, which is even larger in northern water-limited areas (Liu et al., 20119). Incorporating upstream flows and water consumption offers a more reasonable assessment of water stress in the real world. **Some studies have made significant progress in understanding water stress in the YRB by considering upstream components, reservoir operations, or water transfer projects** (Albers et al., 2021; Omer et al., 2020; Xie et al., 2020; Sun et al., 2021). **Yet, they often covered short periods (less than 20 years), thus precluding a comprehensive documentation of the temporal dynamics of water stress.**

References:

Albers, L. T., Schyns, J. F., Booij, M. J., and Zhuo, L.: Blue water footprint caps per sub-catchment to mitigate water scarcity in a large river basin: The case of the Yellow River in China, J. Hydrol., 603, 126992, https://doi.org/10.1016/j.jhydrol.2021.126992, 2021.

Omer, A., Elagib, N. A., Zhuguo, M., Saleem, F., and Mohammed, A.: Water scarcity in the Yellow River Basin under future climate change and human activities, Sci. Total Environ., 749, 141446, https://doi.org/10.1016/j.scitotenv.2020.141446, 2020.

Xie, P., Zhuo, L., Yang, X., Huang, H., Gao, X., and Wu, P.: Spatial-temporal variations in blue and green water resources, water footprints and water scarcities in a large river

basin: A case for the Yellow River basin, J. Hydrol., 590, 125222, https://doi.org/10.1016/j.jhydrol.2020.125222, 2020.

**R2C3:** The manuscript mentions the use of the SWAT model for simulating natural water availability. While the validation against hydrological station data is noted in section 2.4, detailed validation results and statistics (e.g., NSE, R2, P-factor, and R-factor) were not provided in the manuscript. These metrics are important to assess and understand the model's performance comprehensively.

Response: In our previous study (Zhang et al., 2024), we used the Nash–Sutcliffe efficiency (NSE) and the coefficient of determination ($R^2$) to evaluate the performance of the SWAT model. The figure below depicts monthly comparisons between modeled and observed streamflow at 11 hydrological stations during the calibration and validation periods. Generally, the SWAT model performed well in most cases, showing better performance for stations along the main stream than those along the tributaries, indicated by the higher NSE and $R^2$. Specifically, both the NSE and $R^2$ were > 0.7 at five main hydrological stations in both the calibration and validation periods and both the NSE and $R^2$ were > 0.6 (except the Zhuangtou) at five tributary hydrological stations, suggesting that the SWAT model can well capture the temporal variations in streamflow and can be used to simulate the water availability in the Yellow River basin.

We will also include the model performance evaluation in the revised version, which will be provided in the supplementary materials.

[Figure]

**Figure** Comparison between the monthly observed natural streamflow and modeled streamflow in calibration (1965–1975) and validation (1976–1985) periods for 11 hydrological stations. Abbreviation for hydrological stations: TNH = Tangnaihai, LZ = Lanzhou, TDG = Toudaoguai, LM = Longmen, HYK = Huayuankou, ZT = Zhuangtou, ZJS = Zhangjiashan, LJC = Linjiacun, HJ = Hejin, WZ = Wuzhi, and HSG = Heishiguan. NSE$_C$ ($R^2_C$) and NSE$_V$ ($R^2_V$) indicate the Nash-Sutcliffe efficiency values (the coefficient of determination) of the calibration and validation period, respectively.

**R2C4:** In section 2.4 the authors re-run the SWAT model with fixed land use in 1990 but varied climatic conditions to assess the impact of vegetation restoration. By fixing land use to the conditions of 1990, the model controls for the influence of land cover and land use changes. Any changes observed in water availability or WSI in this experiment can thus be attributed solely to climatic variations NOT vegetation restoration, isn't it?

Response: Thanks for your seriousness. We feel sorry for any incomplete descriptions in the original manuscript. In the original study, we run the model using climatic data from 2000 to 2013 and land cover data from 2010 under a normal scenario. To assess the impact of vegetation restoration on water availability, we re-run the model with fixed land cover from 1990, maintaining the same climatic conditions as in the normal scenario (2000–2013). The difference observed can be attributed to the impacts of vegetation restoration. In the revised version, as suggested in Comments 6 and 7, we will try our best to extend the dataset and use the same scenario analysis to quantify the effects of vegetation restoration.

**R2C5:** The introduction section in the study highlights a 120% increase in total water consumption, including both surface and groundwater, in the YRB from the 1960s to 2009. However, upon reviewing the methods and results sections, it is apparent that groundwater pumping and usage were not directly factored into the water availability calculations used in the water scarcity equation. Omitting this factor may lead to an underestimation of water availability and, thus, an overestimation of water scarcity levels.

Response: Thank you for your careful review. **In our water stress assessment, water availability refers to renewable water resources in rivers** (Yin et al., 2017; Wada et al., 2011), including surface, lateral flow, and baseflow, as simulated by the SWAT model. **The baseflow represents water from the shallow aquifer that returns to the reach. Thus, we indirectly considered the impact of groundwater.** Previous studies have reported that (Huang et al., 2021; Veldkamp et al., 2017), the absolute values of water storage in groundwater aquifers are difficult to estimate and often unknown. Consequently, we did not account for these non-renewable water resource components in our assessment. However, as you indicated, this approach may underestimate water availability in regions heavily dependent on groundwater resources, potentially leading to a lower anticipated water stress level in reality. **A more detailed explanation of the calculation of available water resources will be included in the methods and uncertainty sections of the revised version.**

References:

Huang, Z., Yuan, X., and Liu, X.: The key drivers for the changes in global water scarcity: Water withdrawal versus water availability, J. Hydrol., 601, 126658, https://doi.org/10.1016/j.jhydrol.2021.126658, 2021.

Veldkamp, T. I. E., Wada, Y., Aerts, J. C. J. H., Döll, P., Gosling, S. N., Liu, J., Masaki, Y., Oki, T., Ostberg, S., Pokhrel, Y., Satoh, Y., Kim, H., and Ward, P. J.: Water scarcity hotspots travel downstream due to human interventions in the 20th and 21st century, Nat. Commun., 8, 15697, 10.1038/ncomms15697, 2017.

Wada, Y., van Beek, L. P. H., and Bierkens, M. F. P.: Modelling global water stress of the recent past: on the relative importance of trends in water demand and climate variability, Hydrol. Earth Syst. Sci., 15, 3785-3808, 10.5194/hess-15-3785-2011, 2011.

Yin, Y., Tang, Q., Liu, X., and Zhang, X.: Water scarcity under various socio-economic pathways and its potential effects on food production in the Yellow River basin, Hydrol. Earth Syst. Sci., 21, 1-29, 10.5194/hess-21-791-2017, 2017.

**R2C6:** The study's prioritization of water use sectors during future water stress periods aims to mitigate socio-economic impacts by focusing on essential needs. However, this approach is unreliable due to two-sided uncertainties. First, using past period (P4: 2000-2013) water availability to calculate future water deficits ignores the high variability in water availability and the impacts of global climate change, making stationarity an invalid assumption. Second, projecting future water demands based solely on historical trends fails to account for potential changes in socio-economic dynamics and policies, which could significantly alter future demands. While the authors acknowledge the limitations of using P4 water availability and historical water demand trends, I think the resulting water allocation prioritization remains unreliable for policymakers. Addressing uncertainties on at least one side would improve the reliability of the prioritization framework.

Response: Thank you for your insightful comment. As you are concerned, we also observed an overall downward trend in water stress in the study area after 2003,

resulting in slightly higher water stress during the period 2000–2013 compared to the 1990s (1.17 versus 1.12). The Chinese government has implemented more stringent water management policies since 2012, leading to stagnation or even a decrease in water withdrawal in some regions (Huang et al., 2023). According to the latest Water Resources Bulletin of the Yellow River basin (2014–2020), irrigation and industrial water withdrawals show an insignificant and significant ($p<0.05$) decreasing trend, respectively, at the basin scale. Combined with a significant upward trend in domestic water withdrawal, the total water withdrawal has remained relatively constant. As a result, water availability has slightly increased, leading to a decrease in overall water stress in the basin.

At the sub-basin scale, however, we need to collect more detailed human water withdrawal data at finer scale (grid, prefectural, or provincial levels). Moreover, we will collect recent meteorological data and rerun the model to calculate available water and corresponding water stress. **Overall, using the best available information, we aim to extend the study period (1965–2013) and conduct an analysis similar to the current study.** Previous studies in this basin, based on multiple model predictions, **indicate that projected changes in runoff are not significant during the 2000 to 2030 period (Yin et al. 2017; Yin et al. 2020).** Therefore, we mainly focus on the impacts of water use on future water stress. **Meanwhile, instead of the original linear trend forecasting method, we will use the autoregressive integrated moving average (ARIMA) to predict future water demand.** ARIMA is a well-established and effective linear statistical model for time series forecasting that considers both trends and white noise, and is widely used in water demand forecasting (Adamowski et al., 2012; Kavya et al., 2023). We believe that forecasting near-term future water demand (2030s) based on the trajectories of updated water use data with the ARIMA method can significantly reduce uncertainty. **Furthermore, we will collect newly published policies** (e.g., the 14th Five-Year Plan for water resources in various cities and provinces in the Yellow River basin) **to accurately assess the impacts of irrigation efficiency improvements on alleviating future water stress**. We expect that these efforts will enhance the reliability

of this manuscript.

References:

Adamowski, J., Fung Chan, H., Prasher, S. O., Ozga-Zielinski, B., and Sliusarieva, A.: Comparison of multiple linear and nonlinear regression, autoregressive integrated moving average, artificial neural network, and wavelet artificial neural network methods for urban water demand forecasting in Montreal, Canada, Water Resour. Res., 48, https://doi.org/10.1029/2010WR009945, 2012.

Huang, Z., Yuan, X., Liu, X., and Tang, Q.: Growing control of climate change on water scarcity alleviation over northern part of China, Journal of Hydrology: Regional Studies, 46, 101332, https://doi.org/10.1016/j.ejrh.2023.101332, 2023.

Kavya, M., Mathew, A., Shekar, P. R., and P, S.: Short term water demand forecast modelling using artificial intelligence for smart water management, Sustainable Cities and Society, 95, 104610, https://doi.org/10.1016/j.scs.2023.104610, 2023.

Yin, Y., Tang, Q., Liu, X., and Zhang, X.: Water scarcity under various socio-economic pathways and its potential effects on food production in the Yellow River basin, Hydrol. Earth Syst. Sci., 21, 1-29, 10.5194/hess-21-791-2017, 2017.

Yin, Y., Wang, L., Wang, Z., Tang, Q., Piao, S., Chen, D., Xia, J., Conradt, T., Liu, J., Wada, Y., Cai, X., Xie, Z., Duan, Q., Li, X., Zhou, J., and Zhang, J.: Quantifying water scarcity in Northern China within the context of climatic and societal changes and South-to-North Water Diversion, Earth's Future, 8, e2020EF001492, https://doi.org/10.1029/2020EF001492, 2020.

**R2C7:** Additionally, the study's exclusion of the most recent decade (2013-2023) raises concerns. This period has seen significant changes in both water availability and demand, advances in data collection, and new policies and management practices. Incorporating recent data would provide a more accurate and up-to-date assessment of water scarcity in the Yellow River Basin (YRB), reflecting current conditions and offering a better foundation for future projections and management strategies. Including

this recent decade would enhance the study's relevance and accuracy, making it more useful for policymakers.

Response: We acknowledge that the lack of data from the recent decade is a potential limitation of our study. Although your suggestion means a lot to us, **we will certainly make every effort to collect more human water withdrawal and meteorological data to rerun the model and conduct an analysis similar to the current study.** Furthermore, to accurately assess the impacts of irrigation efficiency improvements on alleviating future water stress, we will **incorporate newly published policies (please also refer to the above response).** It should be noted that, to maintain consistency with Zhou et al. (2020) (water withdrawal data used in the current study) and due to data access limitations, we may only be able to obtain water use data categorized by major sectors. Regardless, we believe that conducting the relevant analysis with updated data will provide valuable insights for policymakers.

**R2C8:** The manuscript effectively highlights potential improvements in irrigation efficiency as a key strategy for mitigating future water stress in the Yellow River Basin (YRB). However, it would benefit from a more comprehensive analysis or discussion on the feasibility of achieving these efficiency improvements.

Response: Thank you for your valuable comment. Based on your previous suggestion, **we will collect irrigation water efficiency targets from the latest water use policies of various provinces and cities within the Yellow River basin.** Then, we will further quantify the impact of future irrigation efficiency improvements on alleviating water stress, assuming that the current targets are achievable. In fact, according to previous water resource planning documents released by the Chinese government, these irrigation water efficiency targets are generally attainable (see the table below).

**Table Irrigation efficiency in 2020 (target and actual). The underlined and bold texts indicate the provinces in the Yellow River basin**

| Provinces | Target | Actual | Provinces | Target | Actual |
|---|---|---|---|---|---|
| China | 0.550 | 0.565 | **Henan** | 0.616 | 0.617 |
| Beijing | 0.750 | 0.750 | Hubei | 0.524 | 0.528 |
| Tianjin | 0.720 | 0.720 | Hunan | 0.540 | 0.541 |
| Hebei | 0.675 | 0.675 | Guangdong | 0.500 | 0.514 |
| **Shanxi** | 0.550 | 0.551 | Guangxi | 0.500 | 0.509 |
| **Inner Mongolia** | 0.550 | 0.564 | Hainan | 0.570 | 0.572 |
| Liaoning | 0.592 | 0.592 | Chongqing | 0.500 | 0.504 |
| Jilin | 0.600 | 0.602 | **Sichuan** | 0.480 | 0.484 |
| Heilongjiang | 0.600 | 0.613 | Guizhou | 0.486 | 0.486 |
| Shanghai | 0.738 | 0.738 | Yunnan | 0.492 | 0.492 |
| Jiangsu | 0.600 | 0.616 | Xizang | 0.450 | 0.451 |
| Zhejiang | 0.600 | 0.602 | **Shaanxi** | 0.580 | 0.579 |
| Anhui | 0.535 | 0.551 | **Gansu** | 0.570 | 0.570 |
| Fujian | 0.547 | 0.557 | **Qinghai** | 0.500 | 0.501 |
| Jiangxi | 0.510 | 0.515 | **Ningxia** | 0.530 | 0.551 |
| Shandong | 0.646 | 0.646 | Xinjiang | 0.570 | 0.570 |

Minor Comments

Figures and Tables:

**R2C9:** Figure 1 lacks a legend to explain the various elements used in the diagram. I think clarifying the meaning of the solid and dashed arrows, different rectangular colors, shapes, and outlines would help to understand the content of the figure.

Response: Thanks for your suggestion. We have revised Figure 1 and included the following explanations in the legend to enhance its clarity.

[Figure]

**Figure 1.** Framework for water scarcity assessment. The red, orange, blue, and green colors denote water scarcity assessment, water withdrawal, water availability, and future water deficit, respectively. The rectangle and rounded rectangles denote the main and detailed components of the above four parts, respectively. The dashed and solid arrows denote impact factors and solving measures, respectively.

**R2C10:** Figures 3, 4, and 7 are central to the manuscript's findings but could be clarified. Ensure that all figures have clear legends, labels, and units. Color gradients should be distinct enough for readers to differentiate between categories. Moreover, ensure all figures and tables are referenced in the text and clearly explained. For example, Figure 3 is mentioned, but its significance and interpretation could be better integrated into the discussion.

Response: In the revised manuscript, we will modify the color scheme, units, and labels in Figure 3 to enhance clarity and facilitate better interpretation of the data. Additionally, we will discuss the driving factors behind changes in water stress in the context of recently implemented water resources management policies. To our knowledge, few studies have examined the duration and frequency of water scarcity in this basin. Therefore, we will also discuss the implications and uncertainties associated with these results.

**R2C11:** The use of the terms and definitions: Throughout the manuscript, ensure consistent use of terms and clear definitions. For example, ensure terms like "water scarcity," "water stress," and "water availability" are defined clearly and used consistently to avoid confusion.

Response: Thanks for your seriousness. To avoid confusion, **we will standardize terminology and provide clear definitions**. In this study, we applied a very widely used indicator (WSI) to assess water stress conditions (defined as a ratio between water use and water availability, also see equation 1). A higher WSI value indicates more severe water stress conditions. Consistent with previous research (Veldkamp et al., 2017; He et al., 2020), a WSI value greater than 1 signifies that water resources are insufficient to meet both environmental and human needs, resulting in water scarcity. Water availability in this context encompasses locally generated runoff and incoming discharge from upstream sub-basins, taking into account environmental flow requirements (EFR) and upstream water consumption (Liu et al., 2019).

References:

He, C., Liu, Z., Wu, J., Pan, X., Fang, Z., Li, J., and Bryan, B. A.: Future global urban water scarcity and potential solutions, Nat. Commun., 12, 4667, 10.1038/s41467-021-25026-3, 2021.

Liu, X., Tang, Q., Liu, W., Veldkamp, T. I. E., Boulange, J., Liu, J., Wada, Y., Huang, Z., and Yang, H.: A spatially explicit assessment of growing water stress in China from the past to the future, Earth's Future, 7, 1027-1043, https://doi.org/10.1029/2019EF001181, 2019.

Veldkamp, T. I. E., Wada, Y., Aerts, J. C. J. H., Döll, P., Gosling, S. N., Liu, J., Masaki, Y., Oki, T., Ostberg, S., Pokhrel, Y., Satoh, Y., Kim, H., and Ward, P. J.: Water scarcity hotspots travel downstream due to human interventions in the 20th and 21st century, Nat. Commun., 8, 15697, 10.1038/ncomms15697, 2017.

---

## Author Response (AR1)

To begin with, thank you for giving us the opportunity to submit a revised draft of the manuscript for publication in the Hydrology and Earth System Sciences. Thank the editor and all reviewers' great help with providing such invaluable comments and constructive suggestions to improve our manuscript. We sincerely acknowledge the editor and reviewers in the Acknowledgment section. We considered each comment seriously and revised the manuscript accordingly. The individual comments are addressed in the following response and the manuscript has been revised to accommodate the changes.

Dear Authors,

Both Referees agree on the great interest of your study, but they also suggest a number of clarifications and the refinement of some of your analyses, suggesting major revisions.

In your detailed replies you have already identified the further analyses and explanations to add for addressing the referees' comments and I encourage you to submit a revised manuscript including such next steps, and in particular: adding details on the choice of the method for estimating the environmental flow requirements, clarifying the quantification of the effects of vegetation restoration scenarios and elaborating more on the impact of neglecting groundwater availability.

The additional analyses that will require more work on your side are the need (stressed by both referees) to extend the analyses to the last decade and the application of a more refined and reliable model for predicting the future water demand model: I am sure that such further effort will allow a remarkable improvement of the soundness of your work.

Best wishes,

Elena Toth

Response:

**1. Environmental flow requirements** (please also see our response to R1C3):

Pastor et al. (2014) compared and tested different calculation methods for the estimation of EFR. **They demonstrated that the Variable Monthly Flow (VMF) method was most compatible with actual environmental water requirements,** distinguishing between low-flow (60% of a water resource allocated to EFR), mediate-flow (45%), and high-flow conditions (30%). **Given its performance in the seasonal assessment of water availability (Veldkamp et al., 2017), we therefore adopted the VMF method in our study.** We acknowledge that our results will depend on the chosen method to determine EFR and thus water availability. **However, finding an appropriate EFR method that determines water stress is out of the scope of this study. In the revised version, we have added the reasons why we selected the VMF method and the specific algorithm** (please see lines 189–194).

**2. Effects of vegetation restoration** (see our response to R2C4):

In the revised version, we ran the SWAT model with two simulation scenarios. Under a normal scenario, the model was driven by land cover data from 2015 and climatic data from 2001 to 2020. Another scenario was driven by land cover data from 1990 while maintaining the same climatic conditions as in the normal scenario (2001–2020). The difference observed can be attributed to the impacts of vegetation restoration.

**3. Impact of groundwater on water tress estimation** (see our response of R2C5):

In calculating water availability, **we adopted a method widely used in previous studies, which considers only the water resources in the river** (Yin et al., 2017; Liu et al., 2019; Munia et al., 2020; He et al., 2021), including surface, lateral flow, and baseflow, as simulated by the SWAT model. **The baseflow represents water from the shallow aquifer that returns to the river channel, thereby indirectly accounting for the impact of groundwater. Previous studies have reported that the absolute values of water storage in groundwater aquifers are difficult to estimate and often unknown** (Veldkamp et al., 2017; Huang et al., 2021). Consequently, we did not account for the groundwater pumping from deep aquifers in our assessment. Thus, water availability may be underestimated in regions heavily dependent on such

resources, **potentially leading to a lower anticipated water stress level in reality. We detailed the impact of neglecting groundwater availability on WSI estimation in the uncertainty sections.**

**4. Recent decade data and future prediction** (see our response to R1C6, R2C6, R2C7, and R2C8):

We made every effort to collect more human water withdrawal and meteorological data to rerun the model. Based on the availability of public data, the study period was extended to 2020 and the relevant analysis results and figures were updated accordingly (Figures 3–8). Based on the historical trajectory of irrigation water use and the corrected socio-economic data under different Shared Socioeconomic Pathways (SSPs), we predicted the water demand in the YRB for the 2030s. Furthermore, to accurately assess the impacts of irrigation efficiency improvements on alleviating future water stress, we have incorporated recently published policies related to water resources management (i.e., the 14th Five-Year Plan for water resources for various cities and provinces within the YRB). We believe that these efforts can make the predictions more reliable and provide valuable insights for policymakers.

**Reviewer 1:**

This study addresses the critical issue of water scarcity in the Yellow River Basin, presenting a novel and comprehensive assessment framework for analyzing water shortage. By utilizing a combination of models and statistical data, it explores the spatiotemporal changes in this severely water-deficient river basin over a span of nearly sixty years and provides insightful predictions for the future. The research question, which is a unique and crucial aspect in the field, is clearly defined, the thought process is clear, and the logical chain is complete, resulting in scientifically valuable information and conclusions. The figures are also well-designed. However, before formal publication, I have some suggestions for consideration by the authors:

Response: Thank you for your positive comments and valuable suggestions to improve the quality of our manuscript. We have made extensive modifications to both the manuscript and data to make our results convincing. The detailed point-by-point responses are listed below.

**R1C1:** The abstract and the primary texts are too extended; it's recommended that they be simplified and the main contributions highlighted.

Response: Thank you for this suggestion. We simplified the abstract and main body text to highlight the main findings in the revised version. For example, we reduced the general implications of the results, driving factors of changes in irrigation, and some descriptions of future water use. The number of figures in the full text has been reduced from 10 to 8. Specific details of the modification can be found in the revised version.

**R1C2:** In lines L103-L105, explicitly addressing the deficiencies in previous studies would be beneficial. For instance, what specific challenges do global water stress assessments face? This requires further clarification. Additionally, the decision not to consider upstream inflow and consumption, while not a significant issue in my view, should be explained or referenced to strengthen the paper's argument.

Response: Revised as follows (please also see lines 101–117 in the revised manuscript):

Recently, considering quality requirements, a comprehensive series of assessments of nationwide water stress at multiple temporal and geographic scales has been performed in China (Ma et al., 2020a), which has markedly advanced our understanding of current water stress conditions. **However, upstream inflows and water consumption were typically not considered in most of these assessments. Neglecting upstream water availability leads to an overestimation of downstream water stress** (Munia et al., 2020; Sun et al., 2021). Previous work in China showed that the difference in population affected by severe water stress was 60% with and without consideration of upstream water resources, which is even larger in northern water-limited areas (Liu et al., 20119). **Incorporating upstream flows and water consumption provides a more accurate assessment of water stress in the real world.** Some studies have significantly advanced the understanding of water stress in the YRB by accounting for upstream components, reservoir operations, and water transfer projects (Omer et al., 2020; Xie et al., 2020; Albers et al., 2021; Sun et al., 2021). **Yet, these studies often covered only short periods (less than 20 years), thereby limiting the comprehensive documentation of the temporal dynamics of water stress.**

References:

Albers, L. T., Schyns, J. F., Booij, M. J., and Zhuo, L.: Blue water footprint caps per sub-catchment to mitigate water scarcity in a large river basin: The case of the Yellow River in China, J. Hydrol., 603, 126992, https://doi.org/10.1016/j.jhydrol.2021.126992, 2021.

Liu, X., Tang, Q., Liu, W., Veldkamp, T. I. E., Boulange, J., Liu, J., Wada, Y., Huang, Z., and Yang, H.: A spatially explicit assessment of growing water stress in China from the past to the future, Earth's Future, 7, 1027-1043, https://doi.org/10.1029/2019EF001181, 2019.

Omer, A., Elagib, N. A., Zhuguo, M., Saleem, F., and Mohammed, A.: Water scarcity in the Yellow River Basin under future climate change and human activities, Sci. Total Environ., 749, 141446, https://doi.org/10.1016/j.scitotenv.2020.141446, 2020.

Sun, S., Zhou, X., Liu, H., Jiang, Y., Zhou, H., Zhang, C., and Fu, G.: Unraveling the effect of inter-basin water transfer on reducing water scarcity and its inequality in China, Water Res., 194, 116931, https://doi.org/10.1016/j.watres.2021.116931, 2021.

Xie, P., Zhuo, L., Yang, X., Huang, H., Gao, X., and Wu, P.: Spatial-temporal variations in blue and green water resources, water footprints and water scarcities in a large river basin: A case for the Yellow River basin, J. Hydrol., 590, 125222, https://doi.org/10.1016/j.jhydrol.2020.125222, 2020.

**R1C3:** In line L122 regarding environmental flow requirements, there has been extensive discussion of this flow rate of Yellow River from different perspectives, leading to varied estimates; you need to elaborate on how you evaluated these considerations here.

Response: Yes, we agree with you. As you stated, there are indeed many methods to calculate environmental flow requirements (EFR), such as the Tennant method (Tennant, 1976), the Smakhtin method (Smakhtin et al., 2004), the presumptive standard method (Richter et al., 2012), and the variable monthly flow (VMF) method (Pastor et al., 2014). Liu et al. (2021) reported that the impacts of different EFRs on water stress assessment are substantial in many regions but comparatively minor in areas with intensive human water use, such as Northern China and India. Pastor et al. (2014) compared and tested different calculation methods for the estimation of EFR. **They demonstrated that the Variable Monthly Flow (VMF) method was most compatible with actual environmental water requirements,** distinguishing between low-flow (60% of a water resource allocated to EFR), mediate-flow (45%), and high-flow conditions (30%). **Given its performance in the seasonal assessment of water availability (Veldkamp et al., 2017), we therefore adopted the VMF method in our study.** We acknowledge that our results will depend on the chosen method to determine EFR and thus water availability. **However, finding an appropriate EFR method that determines water stress is out of the scope of this study. In the revised version, we have added the reasons why we selected the VMF method and the specific algorithm** (please see lines 189–194).

References:

Liu, X., Liu, W., Liu, L., Tang, Q., Liu, J., and Yang, H.: Environmental flow requirements largely reshape global surface water scarcity assessment, Environmental Research Letters, 16, 104029, 10.1088/1748-9326/ac27cb, 2021.

Pastor, A. V., Ludwig, F., Biemans, H., Hoff, H., and Kabat, P.: Accounting for environmental flow requirements in global water assessments, Hydrol. Earth Syst. Sci., 18, 5041-5059, 10.5194/hess-18-5041-2014, 2014.

Richter, B. D., Davis, M. M., Apse, C., and Konrad, C.: A presumptive standard for environmental flow protection, River Res. Appl., 28, 1312-1321, https://doi.org/10.1002/rra.1511, 2012.

Smakhtin, V., Revenga, C., and Döll, P.: A pilot global assessment of environmental water requirements and scarcity, Water International, 29, 307-317, 10.1080/02508060408691785, 2004.

Tennant, D. L.: Instream Flow Regimens for Fish, Wildlife, Recreation and Related Environmental Resources, Fisheries, 1, 6-10, https://doi.org/10.1577/1548-8446 (1976).

Veldkamp, T. I. E., Wada, Y., Aerts, J. C. J. H., Döll, P., Gosling, S. N., Liu, J., Masaki, Y., Oki, T., Ostberg, S., Pokhrel, Y., Satoh, Y., Kim, H., and Ward, P. J.: Water scarcity hotspots travel downstream due to human interventions in the 20th and 21st century, Nat. Commun., 8, 15697, 10.1038/ncomms15697, 2017.

**R1C4:** In Figure 2 and the introduction of the study area - clarify how you distinguish between upstream and downstream regions. Or rather, readers expect an understanding of how upstream usage creates pressure on downstream resources within your study area description.

Response: The sub-basin delineation is based on the Digital Elevation Model (DEM) using the SWAT model. Additionally, we used the "Burn in" tool in the SWAT model to ensure that the generated sub-basin reaches align with known stream locations. We have added the spatial distributions of river systems and sub-basins in the Supporting

Information (Fig. S1).

[Figure]

Figure S1. (a) Sub-basins generated by the SWAT model, along with meteorological and hydrological stations for driving and calibrating (validating) the model. (b) Large-scale irrigation district and croplands.

**R1C5:** On the Yellow River, policy-making & unified dispatching already dominate human water use. In your proposed analysis framework emphasizing coordination with upstream, does such coordination significantly impact the basin system?

Response: Yes, you are right. In the Yellow River basin, policy-making and unified dispatching are critical in managing human water use. These measures ensure a coordinated approach to water allocation and use, which is essential given the river's limited and highly variable water resources. In this study, we found that upstream flows were responsible for changes in net water availability in 36%–40% of the sub-basins. In these regions, effective coordination with upstream areas can reduce the frequency and severity of water scarcity and ensure a more reliable water supply for various users. Such coordination can greatly influence the basin system by enhancing water availability, alleviating stress, and bolstering overall resilience to climate change.

**R1C6:** We understand that Zhou's data only went up until 2013. Still, the authors need to find ways to explain that the lack of data from the recent decade will not lead to outdated trends affecting analysis or causing bias in conclusion.

Response: As you noted, we agree that the lack of data from the recent decade is a potential limitation of the study. In the revised version, we have made every effort to

**collect recent data on human water withdrawals and meteorological conditions for the period 2014–2020** in order to rerun the model and conduct an analysis similar to that of the original study. Moreover, the corrected second industrial value added and population (urban and rural) under the five **Shared Socioeconomic Pathways (SSP1–5) during 2021–2040 were used to estimate industrial and domestic water use in the 2030s** (please see section 2.4). Additionally, to accurately assess the impacts of irrigation efficiency improvements on alleviating future water stress, **we collected data on irrigation efficiencies for 2020 and 2025 at the prefectural scale from the 14th Five-Year Plan for water resources for various cities and provinces within the YRB.** It was assumed that irrigation efficiency would continue to increase at the same rate after 2025, allowing us to project irrigation efficiency into the 2030s (please see section 2.5). We believe that these efforts can make the predictions more reliable and provide valuable insights for policymakers. Please also refer to our response to Reviewer 2's comments (Comments 6 and 7).

**R1C7:** During the discussion, it is necessary to include a comparative analysis with previous calculations using other water shortage indicators. This will highlight how the contributions of this assessment can compensate for the shortcomings of previous ones.

Response: We sincerely appreciate the valuable comments. In the current manuscript, we have already compared our findings with other similar studies in the Yellow River basin regarding water stress hotspots regions, future water demand, and drivers of changes in water stress. **As you suggested, we added more case studies in this basin to further strengthen our contributions** (please refer to section 4.1 in the revised version for further details).

**Reviewer 2:**

The manuscript provides an in-depth analysis of water scarcity in the Yellow River Basin (YRB) using an integrated analytical framework. The study spans from 1965 to 2013, focusing on critical indicators like the Water Scarcity Index (WSI), frequency, duration, and exposed population. It also projects future water demand and evaluates potential solutions, particularly improving irrigation efficiency.

General Comments

The study addresses a crucial topic in hydrology and water resources management, particularly for a region as significant as the YRB. Integrating historical data, model simulations, and future projections provides a comprehensive overview. However, several areas could benefit from further clarification and refinement.

Response: We greatly appreciate your professional review of our article. As you noted, several issues need to be addressed. Based on your valuable suggestions, we made the necessary corrections to our current manuscript. The detailed corrections are listed below.

**R2C1:** In the introduction, the authors list the limitations of previous studies regarding water use and water withdrawal estimation, highlighting their implications for water stress assessments. While the study acknowledges these limitations and aims to address them, many of the same uncertainties are reiterated in the uncertainty section (section 4.3). This raises the question: if these limitations remain largely unresolved, why emphasize them in the introduction?

Response: Thanks for your comment. One of the key innovations of this study, which distinguishes it from previous research, is the use of **long-term observed water use data** (Zhou et al., 2020) to evaluate the evolution of water stress from 1965 to 2013. In response to your suggestion (Comment 7), **we extended this dataset at the prefectural level to 2020** by collecting Water Resources Bulletins from various provinces within the YRB. The related analysis has been updated. Further, **we predicted future water demand based on the historical trajectory of irrigation water use and the corrected**

**socio-economic data under different Shared Socioeconomic Pathways (SSPs)** (please see section 2.4). Overall, we believe that including data from the most recent decade can incorporate the effect of water efficiency improvements and current water-saving technologies, thereby making water withdrawal predictions more reliable. Thus, **we have modified the corresponding section on uncertainty analysis.**

**R2C2:** The authors' statements in the introduction about the limitations of previous studies using coarse spatial resolution global water scarcity assessments (e.g., 0.5° × 0.5° level) and neglecting upstream water availability are partly valid. However, there are existing studies that have addressed water scarcity in the Yellow River Basin (YRB) at a higher resolution, considering sub-basin scales and upstream water availability (e.g., Albers et al., 2021; Omer et al., 2020; Xie et al., 2020). Given this and the previous comment, the authors should revise the motivation section of the introduction accordingly.

Response: Thank you for providing these references. We have revised this part, as follows:

A wealth of previous studies in China have explored the general feature of water stress in the YRB at different spatial scales, ranging from provincial or prefectural (Zhao et al., 2015; Huang et al., 2023), to river basin scale (Yin et al., 2020), to sub-basin scale (Zhou et al., 2019; Xu et al., 2022), to grid scales (Zhuo et al., 2016). Recently, considering quality requirements, a comprehensive series of assessments of nationwide water stress at multiple temporal and geographic scales has been performed in China (Ma et al., 2020a), which has markedly advanced our understanding of current water stress conditions. However, upstream inflows and water consumption were typically not considered in most of these assessments. Neglecting upstream water availability leads to an overestimation of downstream water stress (Munia et al., 2020; Sun et al., 2021). Previous work in China showed that the difference in population affected by severe water stress was 60% with and without consideration of upstream water resources, which is even larger in northern water-limited areas (Liu et al., 2019). Incorporating upstream flows and water consumption provides a more accurate

assessment of water stress in the real world. **Some studies have significantly advanced the understanding of water stress in the YRB by accounting for upstream components, reservoir operations, and water transfer projects** (Albers et al., 2021; Omer et al., 2020; Xie et al., 2020; Sun et al., 2021). **Yet, these studies often covered only short periods (less than 20 years), thereby limiting the comprehensive documentation of the temporal dynamics of water stress. More importantly, human water usage estimates in both the historical and future have mostly been based on macroscale socio-economic data, such as gross domestic product and population (Wada et al., 2016a; Yin et al., 2017; Liu et al., 2019). However, changes in water use efficiency, particularly in irrigation, were not considered.**

References:

Albers, L. T., Schyns, J. F., Booij, M. J., and Zhuo, L.: Blue water footprint caps per sub-catchment to mitigate water scarcity in a large river basin: The case of the Yellow River in China, J. Hydrol., 603, 126992, https://doi.org/10.1016/j.jhydrol.2021.126992, 2021.

Omer, A., Elagib, N. A., Zhuguo, M., Saleem, F., and Mohammed, A.: Water scarcity in the Yellow River Basin under future climate change and human activities, Sci. Total Environ., 749, 141446, https://doi.org/10.1016/j.scitotenv.2020.141446, 2020.

Xie, P., Zhuo, L., Yang, X., Huang, H., Gao, X., and Wu, P.: Spatial-temporal variations in blue and green water resources, water footprints and water scarcities in a large river basin: A case for the Yellow River basin, J. Hydrol., 590, 125222, https://doi.org/10.1016/j.jhydrol.2020.125222, 2020.

**R2C3:** The manuscript mentions the use of the SWAT model for simulating natural water availability. While the validation against hydrological station data is noted in section 2.4, detailed validation results and statistics (e.g., NSE, $R^2$, P-factor, and R-factor) were not provided in the manuscript. These metrics are important to assess and understand the model's performance comprehensively.

Response: In our previous study (Zhang et al., 2024), we used the Nash–Sutcliffe

efficiency (NSE) and the coefficient of determination ($R^2$) to evaluate the performance of the SWAT model. Generally, the SWAT model performed well in most cases, with the NSE and $R^2$ exceeding 0.7 for stations along the main stream, and exceeding 0.6 for most stations along the tributaries during the validation period, suggesting that the SWAT model can well capture the temporal variations in streamflow and can be used to simulate water availability in the YRB. The performance of the model is described in the main text, with the corresponding figures provided in the supplementary materials (Fig. S2).

[Figure]

**Figure S2.** Comparison between the monthly natural streamflow and modeled streamflow in calibration (1965–1975) and validation (1976–1985) periods for 11 hydrological stations. Abbreviation for hydrological stations: TNH = Tangnaihai, LZ = Lanzhou, TDG = Toudaoguai, LM = Longmen, HYK = Huayuankou, ZT = Zhuangtou, ZJS = Zhangjiashan, LJC = Linjiacun, HJ = Hejin, WZ = Wuzhi, and HSG = Heishiguan. $NSE_C$ ($R^2_C$) and $NSE_V$ ($R^2_V$) indicate the Nash-Sutcliffe efficiency values (the

coefficient of determination) of the calibration and validation period, respectively.

**R2C4:** In section 2.4 the authors re-run the SWAT model with fixed land use in 1990 but varied climatic conditions to assess the impact of vegetation restoration. By fixing land use to the conditions of 1990, the model controls for the influence of land cover and land use changes. Any changes observed in water availability or WSI in this experiment can thus be attributed solely to climatic variations NOT vegetation restoration, isn't it?

Response: Thanks for your seriousness. We feel sorry for any incomplete descriptions in the original manuscript. In the revised version, we ran the SWAT model with two simulation scenarios. Under a normal scenario, the model was driven by land cover data from 2015 and climatic data from 2001 to 2020. Another scenario was driven by land cover data from 1990, while maintaining the same climatic conditions as in the normal scenario (2001–2020). The difference observed can be attributed to the impacts of vegetation restoration.

**R2C5:** The introduction section in the study highlights a 120% increase in total water consumption, including both surface and groundwater, in the YRB from the 1960s to 2009. However, upon reviewing the methods and results sections, it is apparent that groundwater pumping and usage were not directly factored into the water availability calculations used in the water scarcity equation. Omitting this factor may lead to an underestimation of water availability and, thus, an overestimation of water scarcity levels.

Response: Thank you for your careful review. In calculating water availability, **we adopted a method widely used in previous studies, which considers only the water resources in the river** (Yin et al., 2017; Liu et al., 2019; Munia et al., 2020; He et al., 2021), including surface, lateral flow, and baseflow, as simulated by the SWAT model. **The baseflow represents water from the shallow aquifer that returns to the river channel, thereby indirectly accounting for the impact of groundwater. Previous studies have reported that the absolute values of water storage in groundwater**

**aquifers are difficult to estimate and often unknown** (Veldkamp et al., 2017; Huang et al., 2021). Consequently, we did not account for the groundwater pumping from deep aquifers in our assessment. Thus, water availability may be underestimated in regions heavily dependent on such resources, potentially leading to a lower anticipated water stress level in reality. **However, in the revised version, we calculated only the future surface water deficit under different sectoral water demands. A more detailed explanation of the calculation of available water resources is provided in the methods and uncertainty sections.**

References:

Huang, Z., Yuan, X., and Liu, X.: The key drivers for the changes in global water scarcity: Water withdrawal versus water availability, J. Hydrol., 601, 126658, https://doi.org/10.1016/j.jhydrol.2021.126658, 2021.

Liu, X., Tang, Q., Liu, W., Veldkamp, T. I. E., Boulange, J., Liu, J., Wada, Y., Huang, Z., and Yang, H.: A spatially explicit assessment of growing water stress in China from the past to the future, Earth's Future, 7, 1027-1043, https://doi.org/10.1029/2019EF001181, 2019.

Munia, H. A., Guillaume, J. H. A., Wada, Y., Veldkamp, T., Virkki, V., and Kummu, M.: Future transboundary water stress and its drivers under climate change: a global study, Earth's Future, 8, e2019EF001321, https://doi.org/10.1029/2019EF001321, 2020.

Veldkamp, T. I. E., Wada, Y., Aerts, J. C. J. H., Döll, P., Gosling, S. N., Liu, J., Masaki, Y., Oki, T., Ostberg, S., Pokhrel, Y., Satoh, Y., Kim, H., and Ward, P. J.: Water scarcity hotspots travel downstream due to human interventions in the 20th and 21st century, Nat. Commun., 8, 15697, 10.1038/ncomms15697, 2017.

Yin, Y., Tang, Q., Liu, X., and Zhang, X.: Water scarcity under various socio-economic pathways and its potential effects on food production in the Yellow River basin, Hydrol. Earth Syst. Sci., 21, 1-29, 10.5194/hess-21-791-2017, 2017.

**R2C6:** The study's prioritization of water use sectors during future water stress periods

aims to mitigate socio-economic impacts by focusing on essential needs. However, this approach is unreliable due to two-sided uncertainties. First, using past period (P4: 2000-2013) water availability to calculate future water deficits ignores the high variability in water availability and the impacts of global climate change, making stationarity an invalid assumption. Second, projecting future water demands based solely on historical trends fails to account for potential changes in socio-economic dynamics and policies, which could significantly alter future demands. While the authors acknowledge the limitations of using P4 water availability and historical water demand trends, I think the resulting water allocation prioritization remains unreliable for policymakers. Addressing uncertainties on at least one side would improve the reliability of the prioritization framework.

Response: Thank you for your insightful comment. Based on the collected human water use and meteorological data, **we extended the study period to 2020 and updated the relevant analysis results and figures accordingly in the revised version** (Figures 3–8).

In the future, the **annual irrigation water withdrawal** at the sub-basin scale for the 2030s was estimated **based on historical trajectories (from 2001 to 2020)**. The projected second industrial value added and population (urban and rural) at the grid scale under the five **Shared Socioeconomic Pathways (SSP1–5) during 2020–2040 were used to estimate industrial and domestic water use** in the 2030s. **Statistical data at the prefectural level in 2020** from the Statistical Yearbook of various provinces **were used as the benchmark to correct** the projected industrial value added and population under different SSPs. Meanwhile, to quantify the contributions of improved irrigation efficiency to the surface water deficit and water stress in the future, **we collected data on irrigation efficiencies for 2020 and 2025 (target values) at the prefectural scale from the 14th Five-Year Plan for water resources** for various cities and provinces within the YRB (please see lines 274–306 in section 2.4 and 2.5 for further details). Overall, we believe that including data from the most recent decade can incorporate the effect of water efficiency improvements and current water-saving

technologies, thereby making water withdrawal predictions more reliable and providing valuable insights for policymakers.

**Given the high uncertainty in climate change projections** (Greve et al., 2018), we only focused here on the impact of changes in human water usage on water stress. Our estimation of the future WSI considered only the effects of water withdrawals and overlooked the impacts of climate change. Previous studies have projected that changes in **annual runoff in the YRB would not be evident in the near future during the period from 2000 to the 2030s** (Yin et al. 2017; Yin et al. 2020), **but the long-term and seasonal climatic impacts should be considered in our further research** (please see uncertainty in the new version).

References:

Greve, P., Kahil, T., Mochizuki, J., Schinko, T., Satoh, Y., Burek, P., Fischer, G., Tramberend, S., Burtscher, R., Langan, S., and Wada, Y.: Global assessment of water challenges under uncertainty in water scarcity projections, Nat. Sustain., 1, 486-494, 10.1038/s41893-018-0134-9, 2018.

Yin, Y., Tang, Q., Liu, X., and Zhang, X.: Water scarcity under various socio-economic pathways and its potential effects on food production in the Yellow River basin, Hydrol. Earth Syst. Sci., 21, 1-29, 10.5194/hess-21-791-2017, 2017.

Yin, Y., Wang, L., Wang, Z., Tang, Q., Piao, S., Chen, D., Xia, J., Conradt, T., Liu, J., Wada, Y., Cai, X., Xie, Z., Duan, Q., Li, X., Zhou, J., and Zhang, J.: Quantifying water scarcity in Northern China within the context of climatic and societal changes and South-to-North Water Diversion, Earth's Future, 8, e2020EF001492, https://doi.org/10.1029/2020EF001492, 2020.

**R2C7:** Additionally, the study's exclusion of the most recent decade (2013-2023) raises concerns. This period has seen significant changes in both water availability and demand, advances in data collection, and new policies and management practices. Incorporating recent data would provide a more accurate and up-to-date assessment of water scarcity in the Yellow River Basin (YRB), reflecting current conditions and

offering a better foundation for future projections and management strategies. Including this recent decade would enhance the study's relevance and accuracy, making it more useful for policymakers.

Response: We acknowledge that the lack of data from the recent decade is a potential limitation of our study. **We made every effort to collect more human water withdrawal and meteorological data to rerun the model.** Based on the availability of public data, **the study period was extended to 2020 and the relevant analysis results and figures were updated accordingly.** Furthermore, to accurately assess the impacts of irrigation efficiency improvements on alleviating future water stress, we have **incorporated recently published policies related to water resources management in the revised manuscript** (please also refer to the above response for details). It should be noted that, to maintain consistency with Zhou et al. (2020) (who provided the water withdrawal data used in the current study) and due to data access limitations, we were only able to obtain water use data categorized by major sectors. Nevertheless, we believe that conducting the relevant analysis with updated data can provide valuable insights for policymakers.

**R2C8:** The manuscript effectively highlights potential improvements in irrigation efficiency as a key strategy for mitigating future water stress in the Yellow River Basin (YRB). However, it would benefit from a more comprehensive analysis or discussion on the feasibility of achieving these efficiency improvements.

Response: Thank you for your valuable comment. Based on your previous suggestion, **we collected data on irrigation efficiencies for 2020 and 2025 (target values) at the prefectural scale from the 14th Five-Year Plan for water resources for various cities and provinces within the YRB.** Based on previous similar water resource planning documents released by the Chinese government, these irrigation efficiency targets are generally attainable (see the table below). It thus was assumed that irrigation efficiency would continue to increase at the same rate after 2025, allowing us to project irrigation efficiency into the 2030s (see the relative change in Fig. S3). We further

quantified the contributions of improved irrigation efficiency to the surface water deficit and water stress in the future.

**Table** Irrigation efficiency in 2020 (target and actual). The underlined and bold texts indicate the provinces in the Yellow River basin

| Provinces | Target | Actual | Provinces | Target | Actual |
|---|---|---|---|---|---|
| China | 0.550 | 0.565 | **Henan** | 0.616 | 0.617 |
| Beijing | 0.750 | 0.750 | Hubei | 0.524 | 0.528 |
| Tianjin | 0.720 | 0.720 | Hunan | 0.540 | 0.541 |
| Hebei | 0.675 | 0.675 | Guangdong | 0.500 | 0.514 |
| **Shanxi** | 0.550 | 0.551 | Guangxi | 0.500 | 0.509 |
| **Inner Mongolia** | 0.550 | 0.564 | Hainan | 0.570 | 0.572 |
| Liaoning | 0.592 | 0.592 | Chongqing | 0.500 | 0.504 |
| Jilin | 0.600 | 0.602 | **Sichuan** | 0.480 | 0.484 |
| Heilongjiang | 0.600 | 0.613 | Guizhou | 0.486 | 0.486 |
| Shanghai | 0.738 | 0.738 | Yunnan | 0.492 | 0.492 |
| Jiangsu | 0.600 | 0.616 | Xizang | 0.450 | 0.451 |
| Zhejiang | 0.600 | 0.602 | **Shaanxi** | 0.580 | 0.579 |
| Anhui | 0.535 | 0.551 | **Gansu** | 0.570 | 0.570 |
| Fujian | 0.547 | 0.557 | **Qinghai** | 0.500 | 0.501 |
| Jiangxi | 0.510 | 0.515 | **Ningxia** | 0.530 | 0.551 |
| Shandong | 0.646 | 0.646 | Xinjiang | 0.570 | 0.570 |

[Figure]

**Figure S3.** Relative change in irrigation efficiency from 2020 to the 2030s (%).

Minor Comments

Figures and Tables:

**R2C9:** Figure 1 lacks a legend to explain the various elements used in the diagram. I think clarifying the meaning of the solid and dashed arrows, different rectangular colors, shapes, and outlines would help to understand the content of the figure.

Response: Thanks for your suggestion. We have revised Figure 1 and included the following explanations in the legend to enhance its clarity.

[Figure]

**Figure 1.** Framework for water stress assessment. The red, orange, blue, and green colors indicate water stress assessment, water withdrawal, water availability, and future surface water deficit, respectively. The rectangles and rounded rectangles indicate the main and detailed components of the aforementioned four parts, respectively. The dashed and solid arrows indicate impact factors and mitigation measures, respectively.

**R2C10:** Figures 3, 4, and 7 are central to the manuscript's findings but could be clarified. Ensure that all figures have clear legends, labels, and units. Color gradients should be distinct enough for readers to differentiate between categories. Moreover, ensure all figures and tables are referenced in the text and clearly explained. For example, Figure

3 is mentioned, but its significance and interpretation could be better integrated into the discussion.

Response: In the revised manuscript, we have modified the color scheme, units, and labels in all figures to improve clarity and facilitate more accurate data interpretation. Additionally, based on the updated data, we further elaborated on the implications of the results in the discussion.

However, since 2000, China's State Council and Ministries have implemented a series of laws, regulations, action plans, and technology related to water conservation (Zhao et al., 2015; Zhou et al., 2020), especially the strict Three Red Lines in 2012. This led to a widespread stagnation and even a reduction in human water withdrawals in northern China (Huang et al., 2023), indicating the remarkable success of government-led water conservation efforts. **As a result, human water use is no longer the main reason for the increased WSI in most sub-basins** (Fig. 5b). Thus, a regional deceleration in the rate of WSI increase was observed (41.8% versus 21.4%) and the changes in WSI were driven almost equally by water withdrawals and water availability from P2 to P3 (Fig. 4a), underscoring the important role of climate change in influencing water stress under increasingly stringent water management strategies. Additionally, we found that the frequency and duration of water scarcity in the YRB and some sub-basins exceeded 0.5 and 6 months, respectively (Figs. 4b and S5), indicating that water scarcity occurred more than half the time at the monthly scale, despite WSI being less than 1 at the decadal scale. **These seasonal analyses can provide valuable scientific guidance for the planning of reservoirs and water diversion projects.**

**R2C11:** The use of the terms and definitions: Throughout the manuscript, ensure consistent use of terms and clear definitions. For example, ensure terms like "water scarcity," "water stress," and "water availability" are defined clearly and used consistently to avoid confusion.

Response: Thanks for your seriousness. To avoid confusion, **we standardized**

**terminology and provided clear definitions**. In this study, we applied a very widely used indicator (WSI) to assess water stress conditions (defined as a ratio between water use and water availability, also see equation 1). A higher WSI value indicates more severe water stress conditions. Consistent with previous research (Veldkamp et al., 2017; He et al., 2020), a WSI value greater than 1 signifies that water resources are insufficient to meet both environmental and human needs, resulting in water scarcity. Water availability in this context encompasses locally generated runoff and incoming discharge from upstream sub-basins, taking into account environmental flow requirements (EFR) and upstream water consumption (Liu et al., 2019).

References:

He, C., Liu, Z., Wu, J., Pan, X., Fang, Z., Li, J., and Bryan, B. A.: Future global urban water scarcity and potential solutions, Nat. Commun., 12, 4667, 10.1038/s41467-021-25026-3, 2021.

Liu, X., Tang, Q., Liu, W., Veldkamp, T. I. E., Boulange, J., Liu, J., Wada, Y., Huang, Z., and Yang, H.: A spatially explicit assessment of growing water stress in China from the past to the future, Earth's Future, 7, 1027-1043, https://doi.org/10.1029/2019EF001181, 2019.

Veldkamp, T. I. E., Wada, Y., Aerts, J. C. J. H., Döll, P., Gosling, S. N., Liu, J., Masaki, Y., Oki, T., Ostberg, S., Pokhrel, Y., Satoh, Y., Kim, H., and Ward, P. J.: Water scarcity hotspots travel downstream due to human interventions in the 20th and 21st century, Nat. Commun., 8, 15697, 10.1038/ncomms15697, 2017.

---

## Author Response (AR2)

Dear Authors,

both Referees, that I again thanks warmly for their precious support, are satisfied with your careful revision. Referees #2 has identified some remaining inconsistencies and points to be clarified: once you will have amended and clarified such points, we may proceed with the publication.

Best wishes,

Elena Toth

Response: Thank you for handing our submission and offering us an opportunity to revise it. We have addressed all the comments and provided details of all the changes made to the manuscript. We hope the revised manuscript is now acceptable to you.

The authors have satisfactorily addressed many of my concerns. However, a few areas still require further revision or clarification:

1. The terms "water scarcity" and "water stress" are sometimes used interchangeably, but they have distinct definitions in hydrological contexts. This can cause confusion in sections that discuss water withdrawal, availability, and the Water Stress Index (WSI). Ensure consistent use of these terms throughout the manuscript and provide clear definitions in the introduction or methods sections to avoid confusion.

Response: Thanks for your seriousness again. In the revised version, we provided clear definitions in the method section (see section 2.2). The water stress index (WSI), widely used to assess the water stress intensity, is defined as the ratio of water withdrawal to water availability (Equation 1). **A high WSI value in an area represents high water stress intensity, but not necessarily water scarcity. When the WSI is greater than 1 (WSI>1), water resources cannot sustain environmental or anthropogenic needs and a region is considered to experience water scarcity** (Veldkamp et al., 2017; He et al., 2021). **Water stress is more inclusive and broader concept (see Equations 3 and 4).** In addition to the severity of water stress (WSI), frequency and average duration of water scarcity were also used to describe historical water stress (Veldkamp et al., 2017). Correspondingly, these terms are used consistently throughout the manuscript.

2. In section 3.1, the manuscript discusses the percentage of the population experiencing

water scarcity, moving into or out of scarcity, and facing aggravated or alleviated conditions. The percentages of affected populations vary between sub-sections, and the terms "moving into" and "moving out of" water scarcity could be better defined. It would help to ensure that these terms are consistently used throughout the text to avoid confusion for readers who may struggle to follow the shifts between periods and population dynamics.

Response: Thanks for your comment. We have moved the content of Table 1 from the supporting material (Table S1) into the main text (Table 1), and the specific definitions are as follows.

Table 1. Definitions of different types of population exposed to water scarcity between two periods ($WSI_f$ and $WSI_l$ are WSI values in the former and latter periods, respectively.

| WSI value | Classification |
|---|---|
| $WSI_f$ <1 and $WSI_l$ ≥1 | Moving into water scarcity |
| $WSI_f$ ≥1 and $WSI_l$ <1 | Moving out of water scarcity |
| $WSI_f$ ≥1, $WSI_l$ ≥1, and $WSI_l > WSI_f$ | Aggravation of water scarcity |
| $WSI_f$ ≥1, $WSI_l$ ≥1, and $WSI_l < WSI_f$ | Alleviation of water scarcity |

3. There appear to be inconsistencies in section 3.3 regarding the projection of irrigation water use and its contribution to total water demand in the 2030s. The authors state that "regional total irrigation is projected to decrease by 13.3% in the 2030s compared to the recent two decades (P3)," indicating a notable reduction in irrigation water use. However, later, they report that "total water use in the 2030s is projected to be 34.2 km³, with 56.2% (19.2 km³) contributed by irrigation. These two statements are contradictory. A projected 13.3% decrease in irrigation water use should not result in irrigation contributing more than half of the total water demand. Clarification is needed on how the reduction in irrigation aligns with its large projected share of total water use. Please revise the figures or provide additional explanations to resolve this discrepancy.

Response: The two figures are not contradictory. The irrigation water withdrawal in the P3 is 22.1 km³, which is projected to decrease by 13.3% to 19.2 km³ in the 2030s, i.e., 22.1× (100-13.3)/100=19.2. However, due to the increase in industrial, urban, and

domestic water use, the total mean annual water demand of different SSP is projected to be 34.2 km³ in 2030. The proportion of irrigation is 56.2%, i.e., 19.2/34.2×100%=56.2%.

4. In lines 470–480, the authors state that the future surface water deficit is projected to be 0.6–8.36 km³. However, when discussing irrigation efficiency improvements, they mentioned that the reduction in the surface water deficit would be 6.3 km³. It is unclear whether the 6.3 km³ represents the maximum possible reduction or if it is part of the 0.6–8.36 km³ range. Please clarify the relationship between these figures to ensure consistency in the results.

Response: The 6.3 km$^3$ represents the net surface water deficit in the 2030s after accounting for improvements in irrigation water efficiency, assuming that the water needs of all sectors are fully met. To improve clarity, we have revised the sentence as follows. **When all sectoral water usages need to be fulfilled (8.36 km$^3$), the possible improvement of irrigation efficiency in the future could solve 25% of the water deficit (2.06 km$^3$), leading to a net surface water deficit of 6.3 km$^3$.**

5. The study highlights the effects of vegetation restoration on water availability but does not sufficiently explain how this restoration interacts with other water management practices. In some places, the text suggests that vegetation restoration exacerbates water scarcity, while in others, it appears to be part of broader water-saving efforts. Clarifying the overall impact of restoration efforts in relation to other water-saving measures would enhance the consistency of the discussion.

Response: Previous studies have shown that the impact of vegetation restoration on water availability exhibits significant spatial heterogeneity. It decreases water resources in arid areas (measured as precipitation minus evapotranspiration) but increases water resources in humid regions (Feng et al., 2017; Zan et al., 2024). By scenario simulation, we estimated that vegetation restoration led to a 7.9% reduction in runoff in the YRB between 2001 and 2020, thereby exacerbating water stress. This result is expected, given that most of the YRB is classified as arid or semi-arid. Similar findings have been reported for this basin (Zhang et al., 2018), suggesting that local vegetation restoration

efforts in some regions should be approached with caution to avoid increasing water stress in the YRB. In our future water stress assessment, we considered two aspects: reducing water demand (improvements in irrigation efficiency) and increasing water supply (water transfer projects). These measures have mitigated water stress to a certain extent, counteracting the effects of vegetation restoration.

References:

Feng, H., Zou, B., and Luo, J.: Coverage-dependent amplifiers of vegetation change on global water cycle dynamics, J. Hydrol., 550, 220-229, https://doi.org/10.1016/j.jhydrol.2017.04.056, 2017.

Zan, B., Ge, J., Mu, M., Sun, Q., Luo, X., and Wei, J.: Spatiotemporal inequality in land water availability amplified by global tree restoration, Nature Water, 2, 863-874, 10.1038/s44221-024-00296-5, 2024.

Zhang, S., Yang, Y., McVicar, T., and Yang, D.: An analytical solution for the impact of vegetation changes on hydrological partitioning within the Budyko framework, Water Resour. Res., 54, 10.1002/2017WR022028, 2018.

---

## Author Response (AR3)

Dear Editors,

We would like to express our sincere gratitude to you and the reviewers for your insightful suggestions, which have significantly enhanced the quality of this study. In response, we have made several revisions to the previous version to improve clarity. These include defining the various dimensions of water stress (section 2.2), analyzing the impact of vegetation changes on water stress (section 4.2), assessing the role of improved irrigation efficiency in alleviating future surface water deficit (section 3.3), and refining the phrasing of several other sentences. We hope the revised manuscript is now acceptable to you.